# Multicellular factor analysis of single-cell data for a tissue-centric understanding of disease

Ricardo Omar Ramirez Flores[1]*, Jan David Lanzer[1], Daniel Dimitrov[1], Britta Velten[2], Julio Saez-Rodriguez[1]*

[1]Heidelberg University, Faculty of Medicine, and Heidelberg University Hospital, Institute for Computational Biomedicine, BioQuant, Heidelberg, Germany; [2]Heidelberg University, Centre for Organismal Studies, Centre for Scientific Computing, Heidelberg, Germany

**Abstract** Biomedical single-cell atlases describe disease at the cellular level. However, analysis of this data commonly focuses on cell-type-centric pairwise cross-condition comparisons, disregarding the multicellular nature of disease processes. Here, we propose multicellular factor analysis for the unsupervised analysis of samples from cross-condition single-cell atlases and the identification of multicellular programs associated with disease. Our strategy, which repurposes group factor analysis as implemented in multi-omics factor analysis, incorporates the variation of patient samples across cell-types or other tissue-centric features, such as cell compositions or spatial relationships, and enables the joint analysis of multiple patient cohorts, facilitating the integration of atlases. We applied our framework to a collection of acute and chronic human heart failure atlases and described multicellular processes of cardiac remodeling, independent to cellular compositions and their local organization, that were conserved in independent spatial and bulk transcriptomics datasets. In sum, our framework serves as an exploratory tool for unsupervised analysis of cross-condition single-cell atlases and allows for the integration of the measurements of patient cohorts across distinct data modalities.

*For correspondence:
roramirezf@uni-heidelberg.de
(RORF);
pub.saez@uni-heidelberg.de
(JS-R)

## Editor's evaluation

The authors proposed a computational framework, Multicellular Factor Analysis, which is a fundamental advancement in the factor analysis of cross-condition single-cell atlases. The manuscript convincingly demonstrates the application of Multicellular Factor Analysis to uncover multicellular programs associated with disease processes. This innovative framework not only enables unsupervised analysis of single-cell data but also facilitates integration across patient cohorts, marking a helpful contribution to the exploration of molecular alterations in large-scale cross-condition single-cell atlases.

## Introduction

The availability of cross-condition single-cell transcriptomics atlases profiling the pathological state of different tissues and organs in humans has increased during the last years and will continue to expand in different areas of the biomedical field (***Rood et al., 2022***). In these studies a common objective is to compare the molecular profiles of cell types (i.e. cells that potentially share a developmental origin or lineage) across groups of samples (e.g. patient tissues) over distinct conditions or contexts (e.g. during disease). Differential gene expression analysis is usually performed for this task, in which

the gene expression of each cell type is contrasted across various conditions (*Crowell et al., 2020*; *Squair et al., 2021*). This cell-type-centric approach treats each cell-type-specific alteration in disease independently from each other, ignoring particular gene expression changes of one cell type that may relate to the changes of other cell types, here referred to as multicellular programs. Another limitation of these approaches is that they require a specific definition of cross-condition contrasts a priori. Such definitions could disregard other biological and technical variation factors that influence gene expression across cell types.

A set of novel tissue-centric computational methods for multicellular integration have emerged that are helpful in the definition of multicellular programs associated with clinical covariates of interest (*Jerby-Arnon and Regev, 2022*), and the unsupervised analysis of samples from cross-condition single-cell atlases (*Armingol et al., 2022*; *Mitchel et al., 2022*). These multicellular integration methods are extensions of matrix factorization that aim to reduce the dimensionality of the data while retaining most of the variability. In contrast to classic approaches, such as principal component analysis, these methods are capable of dealing with higher order representations, such as the ones from single-cell data, where a sample is described by a collection of different cell types. A key element of multicellular integration methods is that they first transform cross-condition single-cell data into a multi-view representation, in which each view contains the summarized gene expression profile across cells of the same type for each sample. Unlike multimodal integration, where each cell is represented by a collection of distinct feature modalities (e.g. chromatin accessibility and gene expression) and the objective is to map the features across modalities, in multicellular integration the objective is to measure the variability of samples (e.g. patient tissues) across multiple cell types simultaneously.

Although existing multicellular integration methods can capture coordinated gene expression events across cell types associated with disease from single-cell data, no current framework has been proposed to map these multicellular programs to other complementary data types such as spatial and bulk omics. Spatial data could be used to understand the spatial regulation of multicellular alterations in disease. Moreover, multicellular programs could be used to deconvolute cell-type-specific gene expression alterations in disease from bulk transcriptomics data, complementing current cell-type deconvolution methods that only estimate cell-type compositions of tissues (*Avila Cobos et al., 2020*). This integrative framework would facilitate the meta-analysis of patient samples across technologies.

Here, we show that group factor analysis as implemented in multi-omics factor analysis (MOFA) (*Argelaguet et al., 2020*; *Argelaguet et al., 2018*), can be repurposed in a straightforward manner to perform and extend similar tissue-centric analyses as the ones performed by multicellular integration methods, since it uses similar multi-view data representations and model objectives to create latent spaces. Benefiting from the flexibility of the statistical framework of MOFA, multicellular factor analysis overcomes the limitation of data completeness that some multicellular integration methods enforce (*Armingol et al., 2022*; *Mitchel et al., 2022*), where all samples must contain information in all cell-type views and all cell-type views must contain the same features. In contrast to the aforementioned methods, multicellular factor analysis also provides the unique possibility of jointly analyzing samples of independent studies allowing for meta-analysis. Moreover, multicellular factor analysis is capable of modeling various classes of tissue-centric views including cell-type compositional data, spatial organization patterns, or communication inference scores, generalizing the framework of available methods that only model one class of tissue-level views (i.e. gene expression across cell types).

As a case study, we use a collection of acute (*Kuppe et al., 2022a*) and chronic human heart failure atlases (*Chaffin et al., 2022a*; *Reichart et al., 2022b*), together with a public lupus atlas (*Kang et al., 2018a*). We use multicellular factor analysis for the unsupervised analysis of samples in cross-condition single-cell atlases and the inference of multicellular transcriptional programs associated with technical and biological covariates. We present distinct downstream analyses to relate the inferred multicellular programs to pathway activities and functional cell states. Moreover, we use spatial transcriptomics (ST) to identify the areas in tissues where multicellular disease programs occur. Additionally, we demonstrate the possibility of jointly modeling structural and molecular aspects of tissues leveraging compositions and spatial dependencies of cell types. Finally, we use multicellular factor analysis to meta-analyze single-cell data from multiple patient cohorts to infer multicellular programs that are conserved in independent bulk transcriptomics data. Our analyses represent a flexible multicellular framework that integrates single-cell, spatial, and bulk transcriptomics to analyze cross-condition comparisons to understand tissue alterations during disease. We provide a R package (https://github.

com/saezlab/MOFAcellulaR; *Ramirez Flores, 2023a*) and a python implementation (https://liana-py.readthedocs.io/en/latest/notebooks/mofacellular.html; *Dimitrov, 2023*) to facilitate the application of multicellular factor analysis to cross-condition single-cell atlases.

## Results

## Multicellular factor analysis

The generation of a latent space that captures the variability of patient samples across distinct independent measurements is a task that has been addressed by state-of-the-art multi-omics integration methods established for bulk data. The objective of these methods is to integrate independent collections of features (views) measured in the same samples in an unsupervised manner. Hence, we hypothesized that we could repurpose the statistical framework of these multi-view integration methods, such as MOFA (*Argelaguet et al., 2020*; *Argelaguet et al., 2018*), for a multicellular factor analysis to describe the variability of samples from single-cell data across cell types (*Figure 1*). Based on group factor analysis, as implemented in MOFA, multicellular factor analysis can infer a latent space from a collection of cell-type views that contain the summarized gene expression profile of each cell type per patient (e.g. pseudobulk). The variables that form this latent space can be interpreted as coordinated transcriptional changes occurring in multiple cells, here referred to as multicellular programs, providing a tissue-centric understanding of the analyzed sample. The inferred multicellular programs can be associated with complementary continuous or categorical variables of the analyzed samples to identify coordinated expression changes related to technical or biological variability.

Compared to other multicellular integration methods tailored for the inference of multicellular programs and sample-level unsupervised analysis of single-cell data (*Table 1*), multicellular factor analysis using MOFA allows for a more flexible definition of multi-view integration, since it does not restrict cell-type views to the same features. This flexibility enables the inclusion of additional tissue-level descriptions in the model, e.g., cell-type compositions, spatial relationships, and cell communication inference scores, representing a generalization of current available methods. MOFA's structured regularization enables joint modeling of independent studies making multicellular factor analysis suitable for meta-analysis, a unique feature compared to the aforementioned tissue-centric methods. MOFA's inference strategy enables multicellular factor analysis to deal with missing data: samples can partially or completely miss cell-type views. MOFA models are computationally efficient (*Argelaguet et al., 2020*) making multicellular factor analysis scalable to large-scale cross-condition single-cell atlases. The latent space generated with multicellular factor analysis is interpretable, providing measures of the contribution of each view and feature in the construction of the latent space. Finally, building upon these properties, the cell-type-specific gene weights can be used to generate patient maps helpful in the projection and classification of new samples, and disease signatures that can be mapped to other modalities such as spatial and bulk omics (*Figure 1*).

## Multicellular factor analysis for an unsupervised analysis of samples in single-cell cohorts

To show that multicellular factor analysis can perform an unsupervised multicellular analysis of samples profiled with single-cell or nuclei RNA-seq, we fitted a MOFA model to a cross-condition atlas of human myocardial infarction previously generated by us (*Kuppe et al., 2022a*). This atlas profiles distinct phases of myocardial remodeling after infarction, which is a multicellular compensatory process that involves the coordination of multiple cell types for the maintenance of the heart's function after ischemic injury. After quality control, this dataset contained 27 left ventricle heart single nuclei samples of three tissue conditions across seven cell types previously annotated: myogenic (n=13), fibrotic (n=5), and ischemic (n=9) (*Figure 2A*). The seven cell types, previously profiled and annotated across samples, included cardiomyocytes (CMs), fibroblasts (Fibs), pericytes (PCs), and vascular smooth muscle (vSMCs), endothelial (Endos), myeloid, and lymphoid cells. First, we transformed the single-cell data into a multi-view data representation by generating pseudobulk gene expression profiles for each cell type across samples. For each specific cell-type pseudobulk expression matrix, we selected highly variable genes across samples and filtered out lowly expressed and background genes. We then estimated a shared latent space with six factors.

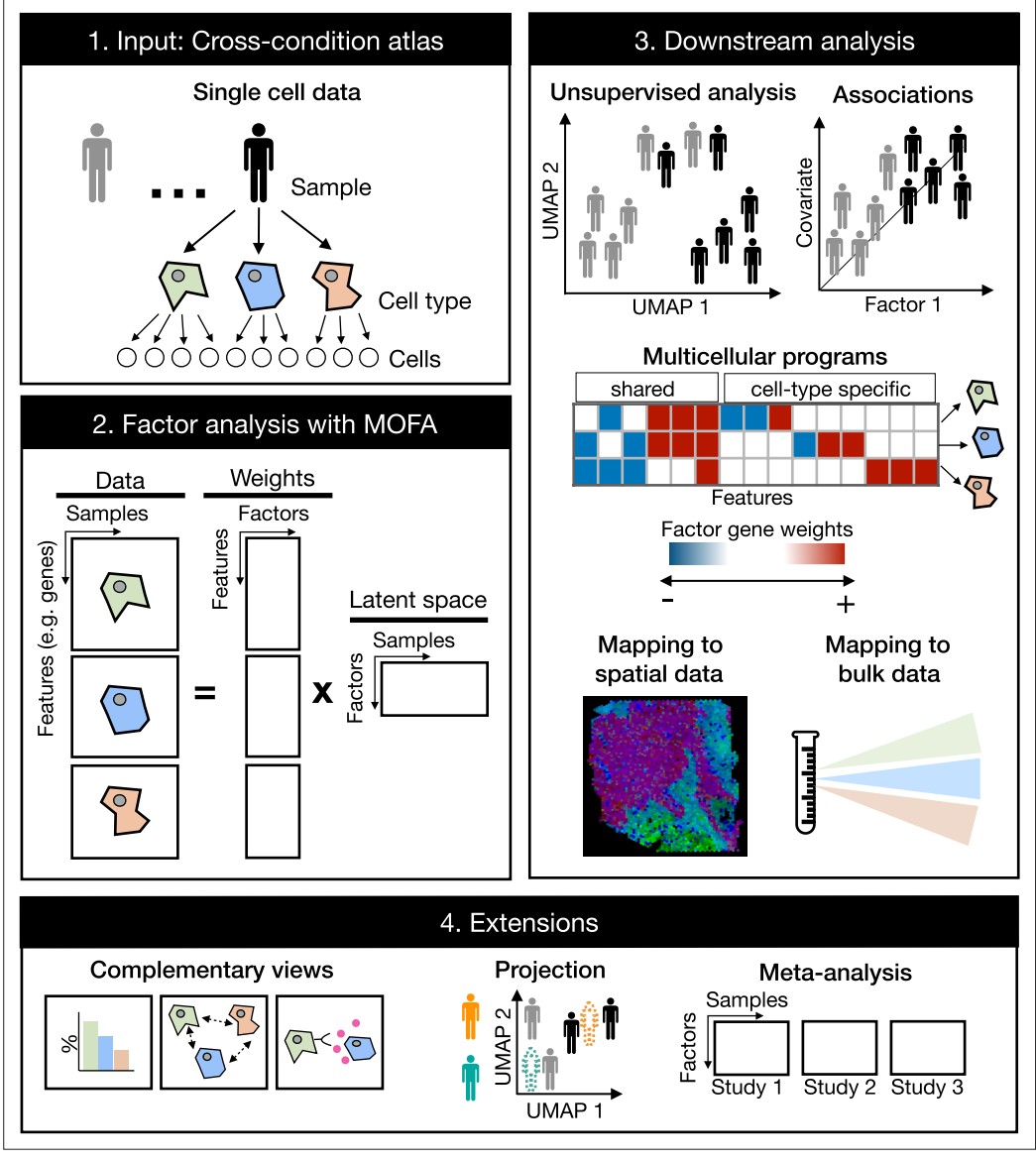

**Figure 1.** Multicellular factor analysis on cross-condition single-cell data. Cross-condition single-cell omics data sample the variability of cells across cell types, patients, and conditions. The information of these datasets can be summarized as a multi-view representation, i.e., a collection of matrices containing cell-type features across samples. Multicellular factor analysis repurposes multi-omics factor analysis (MOFA) to simultaneously decompose the variability of multiple cell types and create a latent space that recovers multicellular transcriptional programs. Throughout this manuscript, several applications are presented to show how this analysis can be used for an unsupervised analysis of single-cell data of multiple samples and conditions, for the identification of multicellular disease processes using the inferred latent space, and for a combined analysis of multiple studies across technologies, such as bulk or spatial transcriptomics. Multicellular factor analysis allows for the inclusion of structural or communication tissue-level views in the inference of multicellular programs, and the joint modeling of independent studies. Moreover, projection of new samples into an inferred multicellular space is also possible.

The latent space returned by the multicellular factor analysis model fitted to the single-cell atlas (*Figure 2A*) explained on average 63.8% of the variability of gene expression of the genes across cell types. Hierarchical clustering of the samples based on their six factor scores effectively separated ischemic, fibrotic, and myogenic-enriched samples. We visualized the sample variability captured by all the factor scores using an Uniform Manifold Approximation and Projection (UMAP) embedding and multidimensional scaling, and observed similar trends of separation of samples from similar conditions (*Figure 2B*, *Figure 2—figure supplement 1A*). From the six recovered factors, Factor 1, 2, and 6

**Table 1.** Comparison of multicellular integration methods.

| Method | Input | | | | | | | Output | | |
|---|---|---|---|---|---|---|---|---|---|---|
| | Statistical method | Decomposition | Data format type | Flexible data type | Multiple groups (e.g. independent studies) | Non-overlapping features across views/layers | Handles missing data | Multicellular programs | Unsupervised analysis of samples | Evaluation of the quality of the latent space |
| DIALOGUE (*Jerby-Arnon and Regev, 2022*) | Penalized matrix decomposition followed by multi-level modeling | Linear | Multi-view | No - only summarized gene expression across cell types or tissue niches | No | Yes | No | Yes | No | No |
| scITD (*Mitchel et al., 2022*) | Tucker decomposition | Linear | Tensor | No - only summarized gene expression across cell types | No | No | No | Yes | Yes | Yes |
| Tensor cell2cell (*Armingol et al., 2022*) | Tensor component analysis | Linear | Tensor | No - only coexpression of ligand and receptors of pairs of sender and receiving cell types | No | No | Yes | Yes | Yes | Yes |
| Multicellular factor analysis with MOFA (*Argelaguet et al., 2020*; *Argelaguet et al., 2018*) | Probabilistic group factor analysis | Linear | Multi-view | Yes - any collection of tissue-level features including summarized gene expression across cell types, cell-type compositional data, cell-type spatial relationships, or cell communication scores | Yes | Yes | Yes | Yes | Yes | Yes |

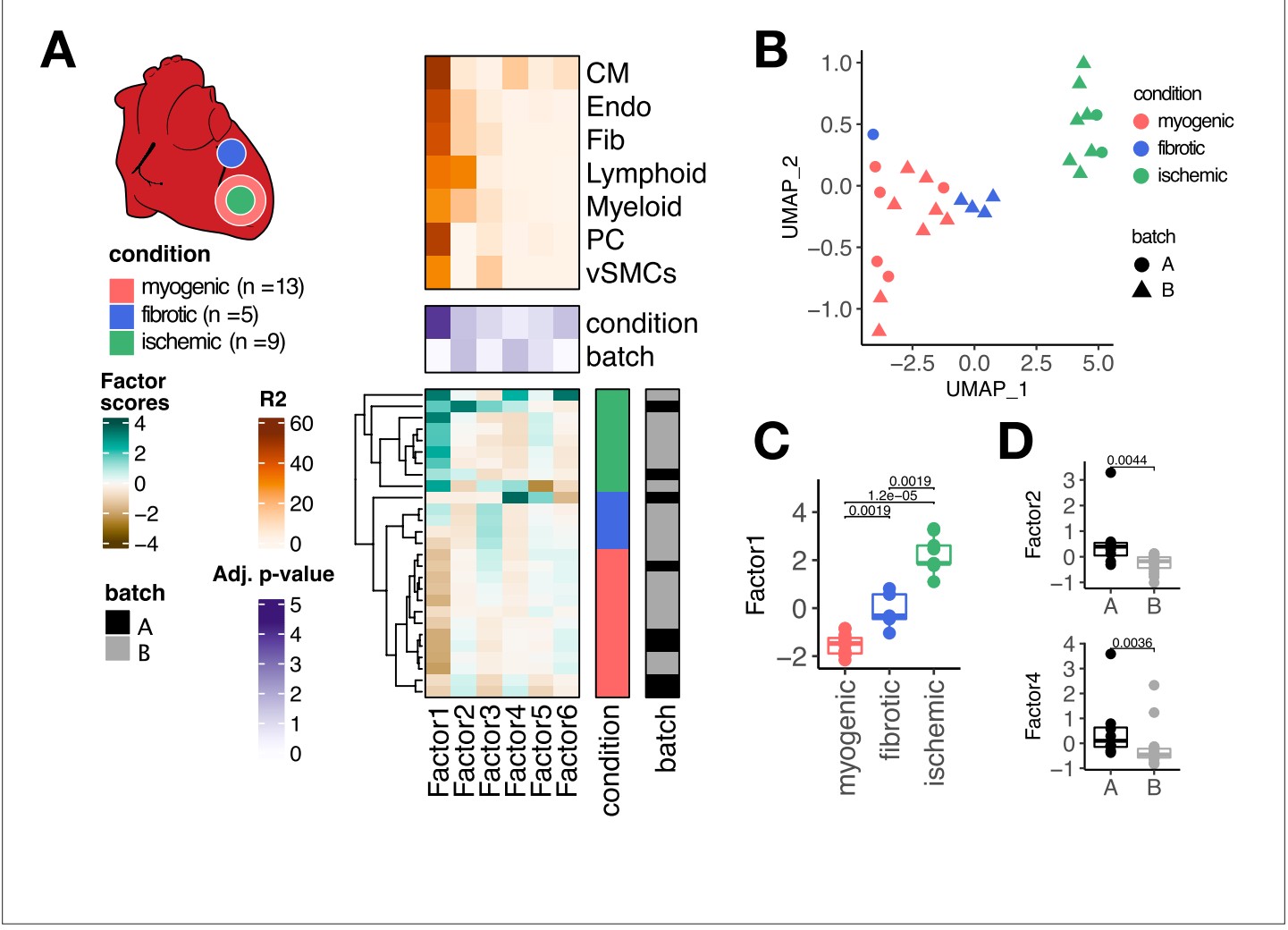

**Figure 2.** Multicellular factor analysis of a single-cell atlas of myocardial infarction. (**A**) Simplified experimental design of a single-cell atlas of acute heart failure following myocardial infarction from *Kuppe et al., 2022a*. The lower panel shows the factor scores of the 27 samples inferred by the model. The condition and technical batch label of each sample are indicated next to each row. Samples are sorted based on hierarchical clustering. The middle panel shows the -log10 (adj. p-values, Kruskal-Wallis test) of testing for associations between the factor scores and the condition (myogenic: n = 13, ischemic: n = 9, fibrotic: n = 5) or batch label. The upper panel shows the percentage of explained variance of each cell-type expression matrix recovered by the factor. (**B**) Uniform Manifold Approximation and Projection (UMAP) embedding of the factor scores of each sample in the acute heart failure atlas. (**C**) Distribution of the scores of Factor 1 across different conditions. (**D**) Distribution of the scores of Factors 2 and 4 across different technical batches. Data information: In (**A–C**), myogenic: n = 13, ischemic: n = 9, fibrotic: n = 5. In (**A, D**), A: n=8, B: n=19. In (**C, D**) data is presented as box plots where the middle line corresponds to the median, the lower and upper hinges correspond to the first and third quartiles, and the whiskers extend no further than 1.5× interquartile range (IQR). Adjusted p-values from Wilcoxon test.

The online version of this article includes the following figure supplement(s) for figure 2:

**Figure supplement 1.** Estimation of a multicellular latent space of acute heart failure using multicellular factor analysis and scITD, and application of multicellular factor analysis for lupus samples.

were associated with the previously defined tissue condition labels (Kruskall-Wallis test adj. p-value <0.05, mean percentage of explained variance across cell types of 52.2%, *Figure 2A and C*), and Factors 2 and 4 were associated with the technical label (Kruskall-Wallis test adj. p-value <0.05, mean percentage of explained variance across cell types of 14.7%, *Figure 2D*). Our results suggest that multicellular factor analysis can be applied to cross-condition single-cell atlases for exploratory unsupervised analysis that enables the detection and prioritization of biological signals.

To evaluate the performance of our proposed multicellular factor analysis in the context of related methods, we compared the latent space inferred by multicellular factor analysis to an analogous one

generated with scITD (*Mitchel et al., 2022*; *Figure 2—figure supplement 1B*) - to our knowledge, the only other tissue-centric method that provides an interpretable latent space to perform both unsupervised analysis of samples and estimation of multicellular programs (*Table 1*). First, we observed that compared to multicellular factor analysis, scITD could only analyze 24 of the 27 samples given the data completeness constraints of their statistical framework based on tensor decomposition. For the shared 24 samples, we evaluated if the latent spaces from both methods could differentiate known labels of patient conditions and technical batches using silhouette scores. Silhouette scores were comparable across methods for all of the biological and technical labels, except for myogenic-enriched samples which were more similar to each other in the multicellular factor analysis's latent space (Wilcoxon test, adj. p-value <0.01, *Figure 2—figure supplement 1C*). Since multicellular factor analysis can handle different sets of genes for each cell-type view, it provides a more flexible framework that enables better control of technical effects in the definition of the latent space, e.g., background genes, compared to methods that enforce data completeness, such as scITD. We quantified the contribution of cell-type-specific marker genes, prone to be background for other cell types, in defining the scITD factor that was associated the most with the patient conditions. We assumed that the definition of the factor would be affected by background noise if TTN, a CM marker gene, and POSTN, a gene expressed in Fibs and Endos, would contain high weights across cell types (*Figure 2—figure supplement 1D*). As expected, scITD's absolute gene weights across cell types were comparable for both marker genes, e.g., POSTN had a high weight in myeloid cells, a clear background effect since POSTN is not expressed by immune cells. Our results show that the statistical framework of MOFA can be repurposed for a multicellular factor analysis of single-cell data that captures the variability of samples across distinct cell types with comparable performance as scITD, the only similar method tailored for this. However, compared to scITD's framework, multicellular factor analysis allows for a more flexible definition of cell-type views that better handles missing information and possible technical biases, such as background gene expression.

To show an additional application of multicellular factor analysis for an exploratory unsupervised analysis of samples profiled with single-cell transcriptomics, we analyzed a peripheral blood mononuclear cell atlas from eight pooled patient lupus samples, each before and after interferon (IFN)-beta stimulation (*Kang et al., 2018a*). After quality control filtering, we analyzed seven cell types with a median number of highly variable genes of 459. A model of four factors explained on average 59% of gene expression variability across cell types. Hierarchical clustering of all factor scores grouped separately stimulated from non-stimulated samples (*Figure 2—figure supplement 1E*). Factor 1, associated with IFN-beta stimulation (Kruskall-Wallis test adj. p-value <0.05), explained on average 50.9% of the variability of gene expression across cell types, being CD14+ monocytes, FCGR3A+ monocytes, and dendritic cells, the cells with the largest explained variance (>60%), suggesting that these cells may be the most responsive to the stimulation.

## Multicellular coordinated programs encoded in the latent space

To characterize the multicellular molecular processes related to myocardial remodeling captured by the latent space inferred with multicellular factor analysis from the human myocardial infarction dataset, we inspected and functionally characterized the cell-type-specific gene weights that defined Factor 1, the factor with the highest association with the sample conditions. As previously mentioned, each factor can be interpreted as higher-order representation of a multicellular program, i.e., coordinated gene expression changes across cell types. These patterns encoded in the gene weights of a factor could include gene expression changes shared across multiple cell types and cell-type-specific expression changes (*Figure 3A*).

First, from the collection of 3136 unique highly variable genes used in the model across cell types, we observed that after filtering by importance (gene weight in Factor 1 different from 0), the median number of genes associated with Factor 1 per cell type was 322 (*Figure 3—figure supplement 1A*). Additionally, 12% of the genes associated with Factor 1 were relevant for more than a single cell type, suggesting that the multicellular coordinated gene expression associated with myocardial remodeling captured by multicellular factor analysis is mainly dominated by cell-type-specific processes (*Figure 3—figure supplement 1B*). To better distinguish between multicellular processes associated with myogenic and ischemic-enriched samples, we simplified the gene weight matrix of Factor 1 into positive and negative cell-type-specific factor gene signatures. Given the positive association

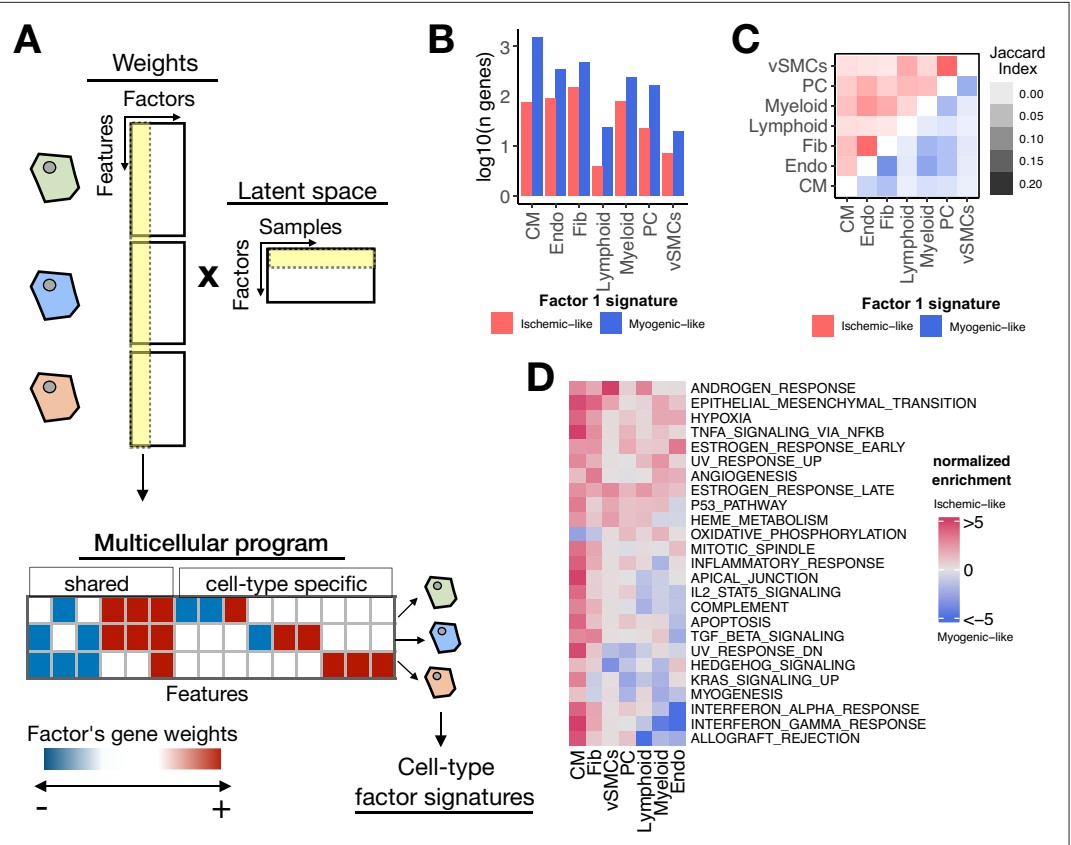

**Figure 3.** Multicellular programs associated with myocardial remodeling. (**A**) Each factor forming the latent space reconstructed by multicellular factor analysis when applied to single-cell data can be interpreted as a higher-level representation of coordinated molecular processes across cell types, here referred to as multicellular programs. The specific cell-type signatures from these programs can be recovered from the feature weights across cell types, where the expression changes of one cell type relate to the other. Moreover, since each multicellular program associates with the variability of samples (e.g. differentially active across conditions), cell-type signatures can also be interpreted in the same context. (**B**) Log10(number of genes) for each cell-type Factor 1 signature associated with the sample conditions of the human myocardial infarction data. Signatures were divided into ischemic-like or myogenic-like signatures based on the weight of each gene. (**C**) Jaccard index across myogenic-like (upper triangle) and ischemic-like (lower triangle) cell-type factor signatures associated with the sample conditions of the human myocardial infarction data. (**D**) Functional enrichment of MSigDB's hallmarks in cell-type signatures. Enrichment is quantified as normalized weighted means, using the gene weights of each cell-type signature. Top 25 pathways based on mean absolute enrichment score are shown.

The online version of this article includes the following figure supplement(s) for figure 3:

**Figure supplement 1.** Coordinated transcriptional programs across cell types in myocardial remodeling inferred with multicellular factor analysis and differential expression analysis.

**Figure supplement 2.** Cell-state-dependent and -independent transcriptional deregulations upon myocardial infarction.

**Figure supplement 3.** Spatial mapping of multicellular programs associated with myocardial remodeling.

of the scores of Factor 1 with ischemic heart samples, the positive and negative cell-type signatures can be understood as ischemic and myogenic signatures, respectively. We observed that cell-type myogenic signatures (median size across cell types=243) were larger than the ischemic ones (median size across cell types=76), indicating a general trend of downregulation of gene expression of the cells in the myocardium after ischemic injury (*Figure 3B*). Additionally, we observed little overlap between myogenic and ischemic cell-type factor gene signatures across cell types (Jaccard index of 0.06 and 0.04, for ischemic and myogenic signatures, respectively, *Figure 3C*). Altogether, our observations suggest that the multicellular transcriptional alterations upon myocardial infarction captured by the

model represent mostly cell-type-specific processes, with a small subset of general processes shared between cell types.

We compared the derived cell-type-specific signatures of Factor 1 with traditional differential expression analysis from pseudobulk expression profiles of tissue samples (*Figure 3—figure supplement 1C*). The median Pearson correlation between the factor gene weights and the log fold changes across cell types was the highest for the contrast between ischemic and myogenic samples (0.98), followed by the contrast between ischemic and fibrotic samples (0.74), and the contrast between fibrotic and myogenic samples (0.65), suggesting that Factor 1 captures the molecular changes associated with the progression of myocardial remodeling, where fibrotic samples represent an intermediary or pseudo-recovered state. Moreover, we observed that from all genes across cell types included in the multicellular program, 77% of them were differentially expressed (edgeR adj. p-value ≤ 0.05) in at least one contrast. In summary, our results suggest a high agreement with traditional differential expression testing, with the advantage that the factor scores and gene weights facilitate the analysis of one condition in the context of all the others, avoiding the need to define multiple contrasts.

Functional characterization of the cell-type-specific myogenic and ischemic factor gene signatures revealed known cellular processes of cardiac remodeling upon myocardial infarction (*Figure 3D*). Enrichment of MSigDB's hallmarks (*Liberzon et al., 2015*) showed that ischemic signatures captured mainly a multicellular response to hypoxia and inflammation across the majority of the cells, together with enrichment of fibrotic processes and angiogenesis. These expected disease processes are associated with tissue damage and cell death upon myocardial infarction which was also captured by the enrichment of the apoptosis pathway in CMs. Myogenic signatures were enriched by homeostatic oxidative phosphorylation processes in CMs and Fibs, together with specific processes of Endos regarding responses to interferons and TGFb activities. Our results suggest that the multicellular programs encoded in the factors provide tissue-level descriptions that facilitate the generation of hypotheses related to disease processes, without the need for independent statistical tests per cell type.

## Cell-type-specific factor gene signatures relate to changes in cell state abundance

We next quantified to what extent the cell-type-specific factor gene signatures recapitulated the emergence of known functional cell states, here defined as cells within cell types with distinct functional phenotypes that do not affect their developmental potential (*Domcke and Shendure, 2023*) (e.g. myofibroblasts). To test for an overrepresented signal of cell states in each cell-type factor signature, we analyzed the enrichment of marker genes of cell states of CMs (n=5), Fibs (n=4), Endos (n=5), and myeloid cells (n=5) presented in our previous work (*Kuppe et al., 2022a*). We observed across cell types that myogenic cell-type factor signatures had an overrepresentation of marker genes of cell states that increased in abundance in myogenic samples, compared to ischemic and fibrotic ones (*Figure 3—figure supplement 2A*, hypergeometric test adj. p-value <0.05). In contrast, ischemic signatures were enriched by marker genes of cell states that increased in abundance in ischemic and fibrotic samples (*Figure 3—figure supplement 2A*, hypergeometric test adj. p-value <0.05). These results align with the expected effect of pseudobulk profiles, where the gene expression signal of the most abundant cells is prioritized. Overall, we showed that cell-type-specific Factor 1 gene signatures captured transcriptional changes related to the change in compositions of functional cell states as a consequence of the disease context.

## Cell-type-specific factor gene signatures are dominated by cell-state-independent transcriptional changes

Next, we questioned if a global transcriptional response to ischemic injury across cells within a cell type could be recovered from the cell-type-specific factor gene signatures. We hypothesized that while the emergence of cell states is a valid abstraction of the molecular processes related to disease, there may be transcriptional changes that are independent from cell states and represent a global alteration of cells within the diseased tissue. This would mean that within a cell type, the deregulation of a gene as a consequence of a disease context can be traced across cell states.

We tested this hypothesis by contrasting the proportion of variance of gene expression that could be explained by the samples' condition and the cell-state classes within each cell-type factor gene

signature (*Lanzer et al., 2023*). Across all cell-type-specific factor gene signatures, we observed differentially expressed genes between conditions (ANOVA adj. p-value <0.01) that were conserved across cell states (*Figure 3—figure supplement 2B, C*). We observed that in general, across all cell-type signatures, a greater proportion of variance of gene expression was explained by the samples' condition, rather than the cell state (one-sample-t-test adj. p-value <0.01, *Figure 3—figure supplement 2D*), suggesting that the genes defining the multicellular latent variable associated with myocardial remodeling recovered by the model capture both cell-state-dependent and -independent transcriptional changes. Moreover, these results suggest that while certain cell states increase in their relative abundance during myocardial infarction, cells within a tissue and cell type partake in a shared global transcriptional response to injury, a novel observation not reported in the original manuscript of this data. These results show the importance of multicellular integration methods for disease description where the focus is to identify coordinated molecular processes across cells in distinct contexts.

## Spatial mapping of multicellular coordinated programs

In addition to the functional characterization of multicellular programs with pathway activities and cell states as presented in the previous sections, complementary data types, such as ST, can be used to better understand their coordination in intact tissues. Thus, we next mapped the cell-type-specific factor signatures associated with myocardial remodeling to the collection of 28 paired ST slides (10× Visium) that were generated together with the single nuclei data used in the previous sections. Given our previous observation that cells within a cell type could respond to cardiac injury in a cell-state-independent manner, we reasoned that the expression of multicellular transcriptional programs associated with myocardial remodeling could be distributed in larger areas in ischemic and fibrotic tissues compared to myogenic-enriched specimens.

For each spatial transcriptomic slide, we calculated the relative area where myogenic and ischemic cell-type factor gene signatures were expressed, using the cell-type composition information in each location. As hypothesized, we observed that across cell types, except for PCs and lymphoid cells, the expression of ischemic programs occur in larger areas and with a bigger magnitude in ischemic samples compared to myogenic samples (Wilcoxon test adj. p-value <0.05, *Figure 3—figure supplement 3A, B*). Similarly, the expression of myogenic programs of Fibs and CMs was more abundant in myogenic samples compared to the ischemic ones (Wilcoxon test adj. p-value <0.05, *Figure 3—figure supplement 3A, B*). Compared to myogenic tissues, fibrotic ones expressed ischemic programs of Fibs in larger areas, while their myogenic programs were expressed in smaller areas (Wilcoxon test adj. p-value <0.05, *Figure 3—figure supplement 3A, B*). These results are in line with the expected disease trajectory of myocardial infarction, which progresses from an acute response to injury to chronic compensation that makes the tissues more similar to a healthy myocardium. Our analyses showed that the multicellular program associated with myocardial remodeling captured by multicellular factor analysis relates to the extent to which myogenic and ischemic cell-type programs are expressed in tissues. Moreover, our mapping strategy provides a complementary analysis strategy for the integration of single-cell and spatial data.

## Multicellular factor analysis for the joint modeling of molecular and tissue-level characteristics of samples

An additional benefit of performing multicellular factor analysis with MOFA is the flexibility to model distinct views with non-overlapping features that enables the incorporation of other tissue-level characteristics in the unsupervised analysis of samples and inference of multicellular programs, such as cell-type compositions and spatial dependencies (i.e. the importance of a cell type in predicting the location and abundance of other cell types) (*Figure 4A*). This modeling alternative distinguishes multicellular factor analysis from available multicellular program inference methods that are limited to a single molecular aspect of tissues, namely gene expression of cell types (*Jerby-Arnon and Regev, 2022*; *Mitchel et al., 2022*) or cell-communication scores (*Armingol et al., 2022*). To showcase the possibility of complementing the inference of multicellular programs with tissue-level descriptions of samples, we extended our previously presented model of human myocardial infarction by including the cell-type compositions of each tissue sample together with spatial dependencies from ST data inferred with MISTy (*Tanevski et al., 2022*; *Figure 4B*). The extended model incorporated four additional sample views. The first of these new views described the compositions of the seven cell-types

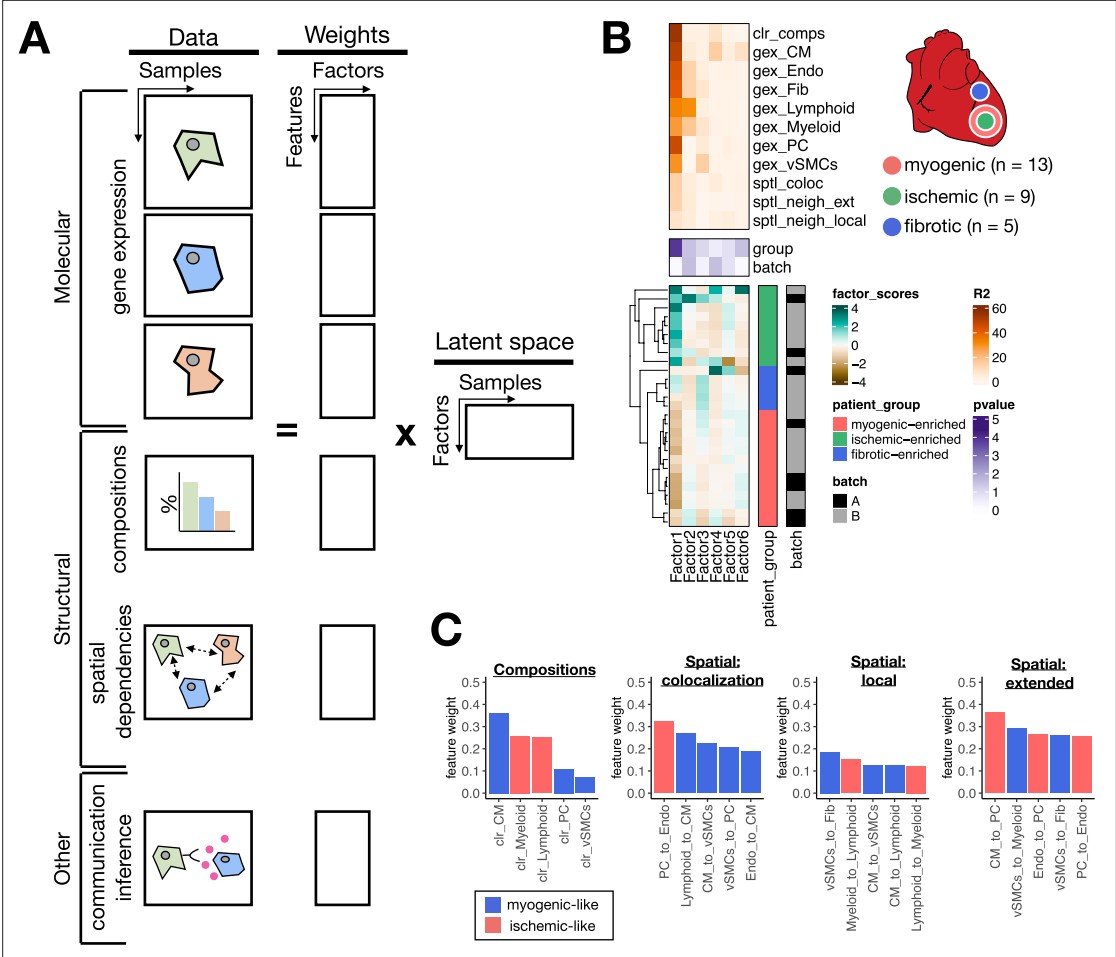

**Figure 4.** Extensions of multicellular factor analysis to model tissue-level molecular and structural features. (**A**) Additionally to molecular views that summarize gene expression across cell types, multicellular factor analysis can model complementary tissue-level features simultaneously, such as cell-type compositions, spatial relationships across cell types, and other functional descriptions such as the co-expression of ligand and receptors between pairs of cells. (**B**) Simplified experimental design of a single-cell atlas of acute heart failure following myocardial infarction from **Kuppe et al., 2022a**. The lower panel shows the factor scores of the 27 samples inferred by the model. The condition and technical batch label of each sample are indicated next to each row. Samples are sorted based on hierarchical clustering. The middle panel shows the -log10(adj. p-values, Kruskal-Wallis test) of testing for associations between the factor scores and the condition (myogenic: n = 13, ischemic: n = 9, fibrotic: n = 5) or batch label. The upper panel shows the percentage of explained variance of each cell-type expression matrix and structural views recovered by the factor. (**C**) Top five feature weights of Factor 1 of the four additional structural views added to the model. Bars are colored based on their weight sign and association with patient groups.

analyzed, and the other three views quantified the spatial dependencies between these seven cell types in three different spatial contexts estimated from ST: colocalization, local-neighborhood, and extended-neighborhood dependencies. The latent space returned by the extended model explained on average 63.8% of the variability of gene expression of the genes across cell types, showing that the extended model did not lose explanatory power of the molecular views of the tissue samples after adding the structural views since the performance was identical to the original model. The factor scores and gene weights across cell types and factors also remained consistent between both models, which could be related to the lower number of features in the additional views. These results suggest that the captured variability of the structural views in the extended model can be related to the coordinated molecular programs associated with myocardial remodeling presented in the past sections.

We observed that the latent space of the extended model captured 70% of the variability in compositions of cell types of the analyzed tissues and on average 22.8% of the variability in spatial dependencies. Feature weights of Factor 1, which associated the most with the sample condition variables (Kruskal-Wallis test adj. p-value <0.05), captured expected changes in cell compositions upon myocardial infarction, particularly the difference between control CM-abundant tissues and

ischemic immune-abundant ones (*Figure 4C*). Moreover, the top five highest feature weights across the spatial dependencies views recovered differential dependencies between immune cells and cells of the vasculature (*Figure 4C*). The low percentage of explained variance captured by the extended model of the spatial dependencies views might suggest that the variability in the spatial organization of cells in cardiac tissues cannot be mainly explained by the patient conditions, and other variables such as the location of tissue sampling may dominate the signal of ST. Moreover, the fact that we could identify shared multicellular programs across samples of the same condition despite variable cellular organization suggests a degree of independence between the local organization of cells in cardiac tissues and their overall response to the ischemic context of myocardial infarction. In sum, we have shown how multicellular factor analysis allows us to relate structural characteristics with molecular changes upon disease.

## Multicellular factor analysis for the meta-analysis of single-cell atlases of heart failure

To show that our proposed framework could be extended to jointly analyze not only multicellular programs and other tissue-level structural or functional features, but also independent patient cohorts, we performed a meta-analysis of publicly available chronic heart failure single nuclei atlases. We created multi-view representations of two different single nuclei studies of heart failure across seven cell types as previously described. The first study (Chaffin2022) encompassed 42 single nuclei cardiac samples profiling healthy myocardium (n=16) and end-stage heart failure both from dilated (n=11) and hypertrophic cardiomyopathies (n=15) (*Chaffin et al., 2022a*). The second study (Reichart2022) profiled 79 cardiac samples of healthy myocardium (n=18) together with samples of dilated (n=52), non-compaction (n=1), and arrhythmogenic right ventricular (n=8) cardiomyopathy (*Reichart et al., 2022b*).

After homogenizing the cell-type annotations, we identified shared highly variable genes per cell type across studies and fitted study-specific models with six factors to define a baseline. Baseline study-specific models captured a mean total amount of explained variance across cell types of 43% for both datasets (*Figure 5—figure supplement 1A, B*). A mean percentage of explained variance across cell types of 25% and 21% was associated with heart failure for Chaffin2022 and Reichart2022, respectively (Kruskal-Wallis test adj p-value <0.05, *Figure 5—figure supplement 1A, B*). For Chaffin2022, we observed additionally that the left ventricle ejection fraction of the patient samples associated with the same factor describing heart failure, as expected (linear model adj p-value <0.05, *Figure 5—figure supplement 1A*). Our results showed that study-specific multicellular programs associated with failing hearts can be inferred using multicellular factor analysis in independent datasets.

Next, we tested if the multicellular programs describing the variability of control and failing myocardium patient samples of each study could be used as reference patient maps where new samples could be projected and classified into a disease condition. First, for each study we generated reference models by training a classifier of healthy and failing myocardium samples from their respective factor scores using random forests (out of bag prediction error of 0.06 and 0.03 for the model of Reichart2022 and Chaffin2022, respectively). Then, we projected the samples of Reichart2022 into the factor space inferred from the samples of Chaffin2022 and vice versa (*Figure 5—figure supplement 1C, D*). Finally, we predicted control or failure labels for projected patient samples using the reference classifier and quantified the performance using precision-recall curves (PRCs). The area under the PRC of the classifier of Reichart2022's patients using Chaffin2022's factors was 0.69, and we observed a higher performance on the classification of Chaffin2022's patient samples using Reichart2022's factors with an area under the PRC of 0.87. These results suggest that the multicellular programs inferred from Reichart2022 better generalize the description of heart failure in comparison to Chaffin2022, which could be explained by the higher degree of variance within the heart failure patients in the former study. Although the generation of patient maps could be useful to compare studies that profile tissue samples of similar phenotypes, a limitation of this approach is that the factors inferred are biased to the variability of the samples used to build the model. Thus, a more robust alternative to find shared multicellular programs across independent studies is to decompose their gene expression variability simultaneously in a single model. Current multicellular integration methods are limited to model a single study, however multicellular factor analysis is directly able to model multiple studies jointly given MOFA's structured regularization for multiple groups of samples (*Argelaguet et al.,*

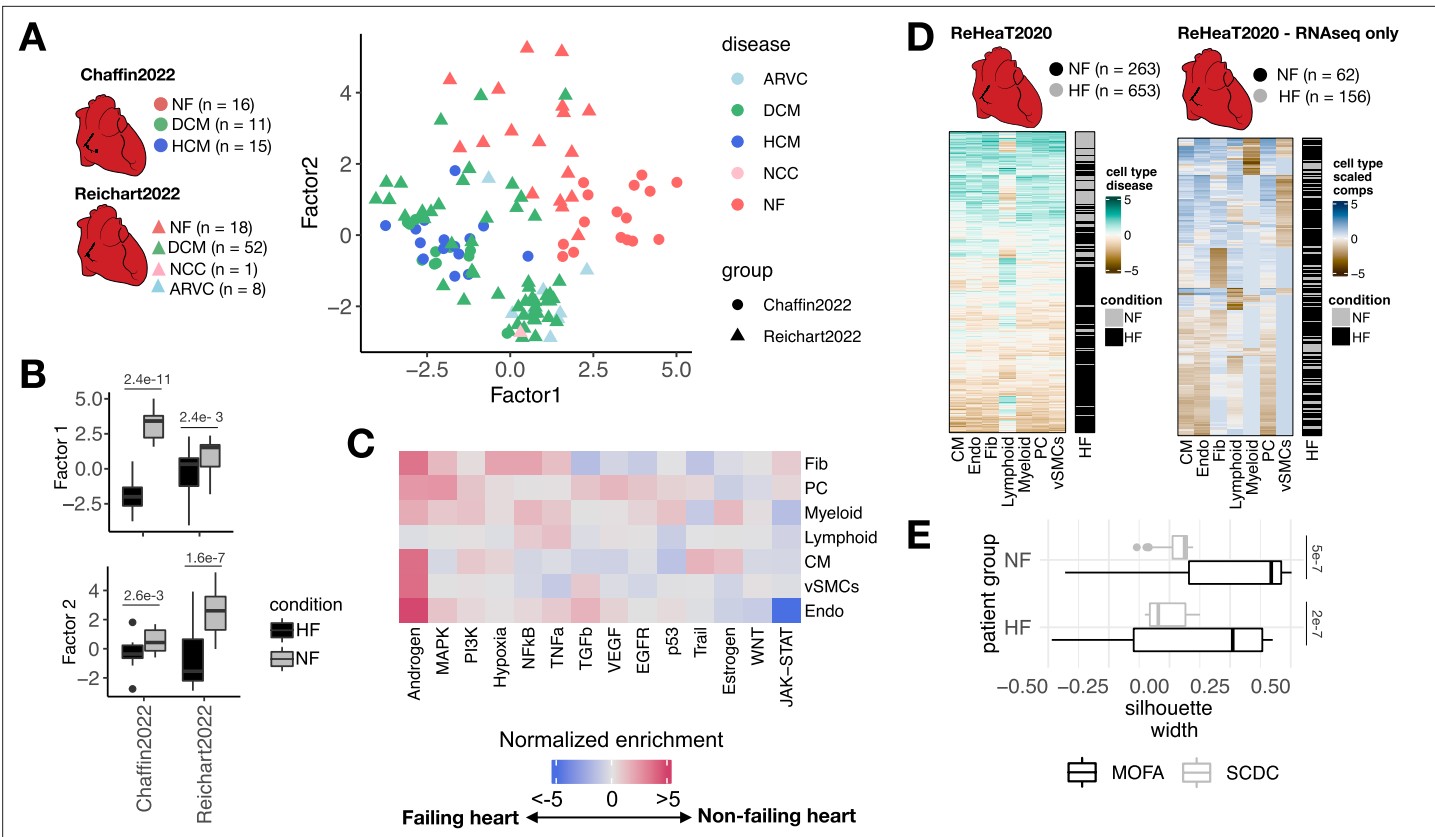

**Figure 5.** Multicellular factor analysis for the meta-analysis of patient cohorts across technologies. (**A**) Distribution of patient samples of two single-cell chronic heart failure studies, Chaffin2022 and Reichart2022, based on the first two factors estimated by a grouped multicellular factor analysis model. (**B**) Distribution of the patient samples of Chaffin2022 and Reichart2022 across the group multicellular Factor 1, separated by their heart failure condition: failing (HF, Chaffin2022: n=26, Reichart2022: n=61) and non-failing (NF, Chaffin2022: n=16, Reichart2022: n=18) hearts. Adjusted p-values from Wilcoxon tests are shown. (**C**) Pathway activities estimated from the gene weight matrices of the group multicellular Factor 1 using PROGENy. CMs = cardiomyocytes, Fibs = fibroblasts, Endos = endothelial cells, vSMCs = vascular smooth muscle cells, PCs = pericytes. (**D**) Hierarchical clustering of bulk transcriptomic samples from ReHeaT using scaled cell-type Factor 1 signatures (left) or scaled cell-type compositions (center-log-ratio transformed) as estimated by SCDC (right). Heart failure status is indicated for each row: failing (HF) and non-failing (NF). (**E**) Distribution of silhouette widths of each RNA-seq sample from ReHeaT grouped by their heart failure. Silhouette widths were calculated either by cell-type Factor 1 signatures (MOFA) or scaled cell-type compositions (center-log-ratio transformed) (SCDC). Adjusted p-values of Wilcoxon tests are shown. HF: n=156, NF: n=62. Data information: In B,E data is presented as box plots where the middle line corresponds to the median, the lower and upper hinges correspond to the first and third quartiles, and the whiskers extend no further than 1.5× interquartile range (IQR).

The online version of this article includes the following figure supplement(s) for figure 5:

**Figure supplement 1.** Multicellular factor analysis to integrate multiple single-cell cohorts and to deconvolute disease signals from bulk transcriptomics.

2020). To show that multicellular factor analysis can be used to infer multicellular programs shared across independent studies, we inferred from Chaffin2022 and Reichart2022 a joint latent space of six factors using MOFA's extension for group modeling and compared them to the baseline study-specific models (*Figure 5—figure supplement 1E*). Our assumption was that the inferred shared latent space would represent multicellular programs that are conserved across distinct etiologies and independent studies. In contrast to the study-specific models, the joint model had a reduction of mean total amount of explained variance across cell types of only 0.6% and 0.51% for Chaffin2022 and Reichart2022, respectively, suggesting that joint modeling had no critical effects on the construction of the multicellular latent space. We visualized the distribution of samples across studies using the scores of the first two factors of the joint model, which suggested a separation of failing and non-failing hearts regardless of their etiology (*Figure 5A*). The joint model had an increased mean percentage of explained variance across cell types associated with heart failure of 4.9% for Chaffin2022 and 9.12% for Reichart2022, supporting the idea that the inclusion of multiple studies could better define conserved

disease signals. From the six factors reconstructed in the joint model, three factors associated with the patient conditions, from which two (Factors 1 and 2) discriminated failing and non-failing hearts in both studies, and two factors (Factor 4) were only associated to the differences between conditions in Reichart2022's (Kruskal-Wallis test adj. p-value <0.05, *Figure 5B*). Functional characterization of the cell-type-specific Factor 1 signatures revealed known multicellular processes active in failing hearts, such as the activation of JAK-STAT, TGFb, and WNT signaling pathways, related to inflammatory and fibrotic processes, together with the reactivation of fetal programs (*Liew and Dzau, 2004*; *Figure 5C*). Our results showed that multicellular factor analysis can be applied to samples coming from distinct patient cohorts for an unsupervised meta-analysis of the transcriptional coordinated responses of a tissue in distinct disease contexts of heart failure. Moreover, the identification of a shared multicellular program of cardiac remodeling associated with heart failure across studies and etiologies suggest the existence of a convergent multicellular functional molecular state of failing myocardium independent of the initial causes of heart failure and sampling variability.

## Mapping multicellular programs to bulk transcriptomics reveals conserved disease signals across technologies

Finally, we proposed an alternative bulk transcriptomics deconvolution approach of disease signals based on our multicellular factor analysis. We assumed that a bulk expression profile is convoluted by both the cell-type compositions and the coordinated multicellular response of all cell types of the tissue and that the contribution of each cell type in the latter effect could be quantified by the enrichment of cell-type-specific factor signatures. To test if heart failure multicellular processes could be traced in independent bulk transcriptomics data, we mapped cell-type-specific heart failure Factor 1 signatures estimated from our previously described joint model of Chaffin2022 and Reichart2022 to an independent collection of 16 bulk heart failure transcriptomics studies (ReHeaT) (*Ramirez Flores et al., 2021*) encompassing 916 human heart samples profiled with microarrays and RNA-seq. Additionally, from the subset of RNA-seq studies in ReHeaT, we estimated cell-type compositions using established bulk deconvolution methods coupled with the heart human cell atlas as a reference (*Litviňuková et al., 2020a*). To justify the selection of the deconvolution method used, we tested the performance of MuSiC (*Wang et al., 2019*), SCDC (*Dong et al., 2021*), and Bisque (*Jew et al., 2020*) in the task of deconvoluting cell compositions from pseudobulk profiles of Chaffin2022 and Reichart2022. We observed that SCDC had the highest median Pearson correlation and the least median root-mean-squared error with the true compositions across studies (median Pearson correlation = 0.84, median root-mean-squared error = 0.108), thus we used this method for the deconvolution of ReHeaT's studies (*Figure 5—figure supplement 1F*).

Hierarchical clustering of cell-type-specific signature scores across samples in ReHeaT showed a general conservation of the heart failure signature identified from single-cell data, in which bulk failing hearts had negative signature scores across cell types (*Figure 5D*, left). We observed that the conservation of the heart failure signal in bulk samples was independent of their estimated cell compositions, since hierarchical clustering of cell-type compositions led to a greater mixing of failing and non-failing samples (*Figure 5D*, right), which was quantified using silhouette scores in only RNA-seq samples (Wilcoxon test, adj. p-value <0.05, *Figure 5E*). Given that the datasets in ReHeaT are of various sample sizes, we then tested if cell-type-specific factor signature scores and cell-type compositions separated failing from non-failing hearts in each study individually. In 9 of the 16 bulk studies, we observed a congruent difference in the expression of at least one cell-type-specific factor signature between failing and non-failing hearts (Wilcoxon test adj. p-value <0.05, *Figure 5—figure supplement 1G*). Additionally, in six of these studies we could differentiate failing and non-failing hearts using all of the cell-type-specific factor signatures except for lymphoid cells (Wilcoxon test adj. p-value <0.05, *Figure 5—figure supplement 1G*). In comparison, differential cell-type compositions of at least one cell type between failing and non-failing hearts were only observed in two of seven RNA-seq studies (Wilcoxon test adj. p-value <0.05, *Figure 5—figure supplement 1G*). These results show that the multicellular responses associated with heart failure estimated from single-cell data are transferable to different patient cohorts and data modalities. The effective mapping of cell-type-specific gene programs in bulk samples suggest that transcriptional profiles from whole tissues are not only driven by highly abundant cell types, e.g., CMs and Fibs in the heart. Moreover, the difference in expression of multicellular programs between failing and non-failing heart samples despite variable cellular

compositions of tissues suggests that disease responses may be independent of local cell-type abundances. Our proposed framework allowed for the meta-analysis of over 1000 human heart samples across scales and technologies, and provides an opportunity to re-analyze bulk data beyond cell-type compositions, serving as a validation ground of single-cell cohorts of smaller size.

## Discussion

Despite the high costs of single-cell technologies, it is expected that in the next few years single-cell datasets encompassing hundreds of patients will be generated. These data hold the promise to enable a better characterization of molecular alterations during disease. Consequently, there is a need for tissue-centric frameworks that on the one hand enable an unsupervised analysis of samples across cell types and on the other hand provide estimations of coordinated molecular programs that better reflect the multicellular nature of organs.

In this study, we propose to repurpose the statistical framework of group factor analysis as implemented in MOFA for a multicellular factor analysis to estimate cross-condition multicellular programs from single-cell transcriptomics data. We demonstrate that the application of multicellular factor analysis to collections of pseudobulk expression matrices of major cell types can generate a latent space that captures technical and biological variability of whole tissue specimens independent of cell-type compositional changes. Multicellular programs can then be applied to build patient maps that allow for the unsupervised analysis of samples, e.g., *Macnair et al., 2022*. Our proposed framework facilitates the simultaneous identification of different cell-type alterations in disease, reducing the number of independent statistical tests and contrasts. The interpretability of the model allows it to prioritize shared coordinated transcriptional changes between cell types, without losing the possibility of identifying cell-type-specific alterations. Additionally, the reconstruction metrics provided by the model can be used to identify subsets of cell types that contribute more to specific clinical covariates of samples. MOFA uses automatic relevance determination (*Argelaguet et al., 2018*) to identify the optimal number of factors forming the latent space, which also facilitates the use of multicellular factor analysis. We argue that, in comparison to novel methods explicitly built for the modeling of multicellular responses (*Armingol et al., 2022*; *Jerby-Arnon and Regev, 2022*; *Mitchel et al., 2022*), multicellular factor analysis has three distinct advantages: (1) it enables to better characterize cell-type-specific responses and to deal with the technical limitations of cell capture and background noise by not enforcing data completeness across samples and cell-type views, (2) flexible view definition with non-overlapping features that allows for extending the model to include molecular and tissue-level descriptions of tissues, as a generalization of available methods, and (3) joint modeling of independent studies to generate a shared latent space for samples, which facilitates the integration, comparison, and meta-analysis of multiple patient cohorts.

In an application to a collection of public single-cell atlases of acute and chronic heart failure, we found evidence of dominant cell-state-independent transcriptional deregulation of cell types upon myocardial infarction not found by previous analyses. This may suggest that while certain functional states within a cell type are more favored in a disease context, most of the cells of a specific type have a shared transcriptional profile in disease tissues. If part of this shared transcriptional profile is interpreted as a signature of the tissue microenvironment that drives cells in tissues toward specific functions, this result may also indicate that a major source of variability across tissues, besides cellular composition, is the degree in which the homeostatic transcriptional balance of the tissue is disturbed. By combining the results of multicellular factor analysis with ST datasets, we explored this hypothesis and identified larger areas of cell-type-specific transcriptional alterations in diseased tissues. Moreover, extending our multicellular factor analysis model with the spatial relationships across cell types revealed a degree of independence between the activation of myocardial remodeling programs and the local organization of cells in the tissue, a finding not reported in the original manuscript of the analyzed dataset or elsewhere. Given these observations on global alterations upon myocardial infarction, we meta-analyzed single-cell samples from two additional studies of healthy and heart failure patients with multiple cardiomyopathies. Here, we found a conserved transcriptional response across cell types in failing hearts, despite technical and clinical variability between patients. Further, we could find traces of these cell-type alterations in bulk datasets that were independent to the cellular compositions of tissues. These observations suggest that our approach can estimate cell-type-specific transcriptional changes from bulk data that, together with changes in cell-type compositions, describe

tissue pathophysiology. Altogether, these results highlight how multicellular factor analysis can be used to integrate the measurements of independent single-cell, spatial, and bulk datasets to measure cell-type alterations in disease.

Our work has a number of limitations. Our proposed framework is dependent on the summarization of the nested design of single-cell data studies using pseudobulk profiles per cell type, which requires the definition of cell-type ontologies before performing a multicellular analysis, an ongoing effort in the single cell community (*Osumi-Sutherland et al., 2021*). In addition, pseudobulk profiles lead to an information loss since the information of multiple cells is aggregated. However, our observations on the conservation of global responses to disease within cell types across scales suggest evidence that current pseudobulk approaches still provide a meaningful understanding of tissue function. Furthermore, the linear constraints of the inferred latent space by group factor analysis restrict the type of gene interactions captured by the model. These limitations, however, are shared across current tissue-centric tailored methods. In contrast, models based on generative deep learning (*Boyeau et al., 2022*; *De Donno et al., 2023*) and the Wasserstein metric (*Chen et al., 2020*; *Joodaki et al., 2022*) can take advantage of single-cell measurements to estimate sample-level heterogeneity, but the interpretability of their estimated latent space is limited in comparison to multicellular factor analysis, where features and cell types can be associated with each factor.

While our proposed approach enables the inference of tissue-level coordinated responses across cell types in distinct contexts, the connection of these processes to cell-cell communication events remains an open challenge. Applications of group factor analysis with MOFA including views measuring the co-expression of ligands and receptors from pairs or groups of cells to infer cell-cell communication programs are possible, analogous to the work of *Armingol et al., 2022*; *Baghdassarian et al., 2023*, as shown in the tutorials of our cell-cell communication tool LIANA+ (*Dimitrov et al., 2023*) (https://liana-py.readthedocs.io/en/latest/notebooks/mofatalk.html). Alternatively, the estimation of multicellular programs could be further used to inform the inference of mechanistic network models connecting inter- and intra-cellular signaling events. However, these approaches are limited by the potential of transcriptomics measurements in explaining cell-cell communication.

Although this study focused on applying group factor analysis using MOFA to understand multicellular responses in tissues, our results also support the application of similar multi-view models to single-cell data, such as MEFISTO (*Velten et al., 2022*) and MuVI (*Qoku and Buettner, 2022*). MEFISTO, which analyzes complex time course experimental designs, could be used to explore multicellular coordinated developmental processes. Additionally, MuVI could improve the interpretation of multicellular coordinated processes by incorporating prior knowledge in the inference of the latent space.

In summary, we contributed with a framework that allows the integration of measurements of independent single-cell, spatial, and bulk datasets to contextualize multicellular responses in disease. We provided an R package and a python implementation within LIANA (*Dimitrov et al., 2022*) to apply multicellular factor analysis in https://github.com/saezlab/MOFAcellulaR (*Ramirez Flores, 2023a*) and https://liana-py.readthedocs.io/en/latest/notebooks/mofacellular.html (*Dimitrov, 2023*), respectively. Our proposed tissue-centric exploratory analysis is scalable and broadly applicable to any single-cell study profiling multiple samples, and it is not limited to transcriptomics measurements or case-control designs.

## Materials and methods
### Multicellular factor analysis
We repurposed the statistical framework of MOFA (*Argelaguet et al., 2020*; *Argelaguet et al., 2018*) to analyze cross-condition single-cell atlases. These atlases profile molecular readouts (e.g. gene expression) of individual cells per sample, following their classification into groups based on lineage (cell types) or functions (cell states). We assumed that this nested design could be represented as a multi-view dataset of a collection of patients, where each individual view contains the summarized information of all the features of a cell type per patient (e.g. pseudobulk). In this data representation, there can be as many views as cell types in the original atlas. MOFA is then used to estimate a latent space that captures the variability of patients across the distinct cell types. The estimated factors composing the latent space can be interpreted as a collection of multicellular programs that capture

coordinated expression patterns of distinct cell types. The cell-type-specific gene expression patterns can be retrieved from the factor loadings, where each gene of each cell type would contain a weight that contributes to the factor score. Similarly, as in the application of MOFA to multi-omics data, the factors can be used for an unsupervised analysis of samples or can be associated with biological or technical covariates of the original samples. Additionally, the reconstruction errors per view and factor can be used to prioritize cell types associated with covariates of interest.

## Datasets

We applied a multicellular factor analysis to three independent published single-cell atlases of acute and chronic heart failure. To ensure the comparability of the analysis across atlases, we defined a heart cell ontology that included the following cell types: CMs, Fibs, Endos, PCs, vSMCs, myeloid and lymphoid cells.

### Human myocardial infarction

Single nuclei RNA-seq (sn-RNA-seq) gene count expression matrices from 27 human heart tissue samples (patient area) from our previous work (*Kuppe et al., 2022b*; data ref: *Kuppe et al., 2022a*) were used. The data was downloaded from the Human Cell Atlas (https://data.humancellatlas.org/explore/projects/e9f36305-d857-44a3-93f0-df4e6007dc97) and imported into a *SummarizedExperiment v1.24.0* R object. We used the provided cell-type annotations. Data from adipocytes, neuronal, and proliferating cells were excluded since they were present in fewer than 26 patients. Samples were previously annotated as myogenic-enriched, ischemic-enriched, and fibrotic-enriched, summarizing the distinct physiopathological zones and time-points after human myocardial infarction.

For validation of the relevance of the multicellular factor analysis applied to this dataset, we used the matching 28 ST slides (10× Visium) provided in the publication. Log-normalized data was generated with *normalize_total* and *log1p* functions from *scanpy v1.9.1* (*Wolf et al., 2018*). Cell-type deconvolution scores per location, previously computed by cell2location (*Kleshchevnikov et al., 2022*), were used as provided in the Human Cell Atlas entry previously mentioned.

### Human heart failure caused by dilated and hypertrophic cardiomyopathies (Chaffin2022)

Gene count expression matrices from 42 sn-RNAseq left ventricle cardiac samples profiling healthy myocardium (n=16) and end-stage heart failure both from dilated (n=11) and hypertrophic cardiomyopathies (n=15) were obtained from *Chaffin et al., 2022a*; data ref: *Chaffin et al., 2022b*. Data was downloaded from https://singlecell.broadinstitute.org/single_cell/study/SCP1303/single-nuclei-profiling-of-human-dilated-and-hypertrophic-cardiomyopathy. Cell-type annotations were aligned to our proposed cell-type ontology using regular expressions. Unannotated cells were discarded.

### Human heart failure caused by dilated and arrhythmogenic cardiomyopathies (Reichart2022)

sn-RNAseq gene count matrices from 79 cardiac samples of healthy myocardium (n=18), together with samples of dilated (n=52), non-compaction (n=1), and arrhythmogenic right ventricular (n=8) cardiomyopathy were collected from *Reichart et al., 2022a*; data ref: *Reichart et al., 2022b*. Left ventricle data of single nuclei samples were selected from the cellxgene entry: https://cellxgene.cziscience.com/collections/e75342a8-0f3b-4ec5-8ee1-245a23e0f7cb/private. Cell-type annotations from the authors were adapted to our ontologies using regular expressions and unannotated cells were discarded. Ensembl IDs used in the count matrix were transformed into gene symbols using bioMart v2.50.3 (*Durinck et al., 2009*) and duplicated entries were summed together.

### Human peripheral blood mononuclear cells from lupus patients

Demultiplexed scRNA-seq count matrices from eight pooled lupus patients samples, each before and after IFN-beta stimulation (*Kang et al., 2018a*, data ref: *Kang et al., 2018b*) were downloaded using pertpy v.0.4.0 (https://github.com/theislab/pertpy; *Heumos and Lotfollahi, 2021*). Cell types used for the analysis were: B cells (B), CD14 positive (CD14) and FCGR3A positive (FGR3) monocytes, CD4 and CD8 T cells (CD4T, CD8T), dendritic cells (DCs), and natural killer cells (NK).

## Creation of pseudobulk expression profiles for multicellular factor analysis

Pseudobulk expression profiles were generated for each major cell type of each independent sample collected in every atlas by summing up the UMI counts of all cells belonging to each of the cell types defined in our ontology. Pseudobulk profiles generated with less than 25 cells were discarded. Genes with less than a minimum of 100 counts in a single sample or detected in less than 25% of the samples were discarded. For the human lupus atlas, genes with less than a minimum of 10 counts in a single sample were discarded. Data was normalized using the trimmed-mean of M values method in edgeR v3.36.0 (*Robinson et al., 2010*) with a scale factor of 1 million and log-transformed. Within each atlas, for each cell-type expression matrix, we selected highly variable genes with two strategies. Highly variable genes across samples in the human myocardial infarction atlas were selected for each cell type using scITD's adaptation of PAGODA2's method (norm_variances >1.5) (*Mitchel et al., 2022*). This was done to enable the comparison between multicellular factor analysis and scITD. In both of the chronic heart failure atlases and the lupus atlas, we identified highly variable genes per cell type using *scran's v1.22.1* (*Lun et al., 2016*) *modelGeneVar* function with a biological variance threshold of 0.

## Exclusion of background genes from pseudobulk profiles

To avoid including genes belonging to background counts of cell-free mRNA in the cell-type views used in the multicellular factor analysis, we limited the genes that could be considered highly variable within each cell type. For each cell type, we filtered out all highly variable genes that could be used as markers for any other cell type. Marker genes of the cell type from which background genes were filtered out were not considered in the procedure. This filtering procedure reduces the chances of including highly expressed cell-type marker genes that would be more likely to be part of the background counts of all the pseudobulk expression profiles.

In all heart datasets we identified cell-type marker genes from the differential expression analysis of cell-type pseudobulk expression profiles using *edgeR v3.36.0* (*Robinson et al., 2010*). Genes with a false discovery rate <0.01 and a log fold change greater than 1 were considered marker genes. Each cell type was compared against the rest in the model design.

## Definition of multicellular factor analysis models for individual single-cell atlases

A MOFA model with six factors was fitted to the collection of pseudobulk cell-type expression matrices, where each cell type represented an independent view, for the acute and chronic heart failure datasets. Gaussian likelihoods were used for each view. Feature-wise sparsity was not forced in the model to obtain the greatest number of genes per cell type associated with each factor. View data was centered before fitting the model. The six factors were recommended while fitting the model using *MOFA2 v1.4.0 run_mofa* function to the human myocardial infarction data. MOFA uses Automatic Relevance Determination (*Argelaguet et al., 2018*) to identify the optimal number of factors forming the latent space. For consistency we kept the same number of factors for the rest of the models. For the lupus dataset *MOFA2 v1.4.0 run_mofa* function recommended four factors.

## Association of multicellular factor scores to covariates

For a given multicellular factor analysis model, we associated the factor scores of each patient sample to reported biological and technical covariates in the dataset using Kruskal-Wallis tests. p-Values were corrected using the Benjamini-Hochberg procedure. In the human infarction atlas, the patient group and the technical batch label were tested for association with factor scores. In the chronic heart failure atlases, the patient's condition, etiology, age, ejection fraction, genotype, and sex were tested for association with factor scores when data were available.

## Fitting an scITD model to human myocardial infarction data

To evaluate the capacity of MOFA to fit a multicellular factor analysis, we compared the latent space inferred from the human myocardial infarction dataset to the one recovered by scITD (*Mitchel et al., 2022*), a related method. To fit a scITD model, first, pseudobulk profiles for each cell type across patients were generated as previously described. Compared to MOFA, scITD represents distinct cell-type views in a tensor, enforcing each cell-type view to contain the same features. Thus, the union

of all highly variable genes across cell types were used in each tensor layer. We used the identical collection of highly variable genes per cell type previously selected for the multicellular factor analysis model. Then, a Tucker decomposition (*Tucker, 1966*) of the pseudobulk tensor was performed with *scITD's v1.0.2 run_tucker_ica* function that discarded patient samples with incomplete profiles. A latent space of six factors was recovered to keep consistency with MOFA's model. Tests for association of inferred factors with clinical covariates were performed with ANOVAs as previously described.

To compare the multicellular latent spaces inferred by multicellular factor analysis and scITD, we evaluated their ability to differentiate pre-defined histological patient groups and technical batches reported in the human myocardial infarction data. Silhouette scores of each sample were calculated relative to their reported histological class (myogenic, fibrotic, or ischemic) and technical batch from an Euclidean distance matrix calculated from either the multicellular factor analysis or scITD factor scores. We performed Wilcoxon tests to compare the silhouette scores for each histological and technical batch class. p-Values were corrected using the Benjamini-Hochberg procedure.

## Definition of cell-type-specific factor signatures from the multicellular factor analysis models

After identifying the multicellular factor that associated the most with a covariate of interest (e.g. differences across sample conditions), we defined two factor gene signatures that capture the positive and negative trends of the factor for each cell type. Given the linear nature of the factor analysis implemented in MOFA, for a cell type of interest, it is possible to separate the genes with a weight different from 0 into two main classes, positive (>0) and negative genes (<0). The interpretation of these two gene sets depends on the direction of association between the factor of interest and the samples' covariates. For example, if a factor X is associated positively with a disease condition, then all genes with a positive weight in a cell type are also associated positively with the disease condition. All cell-type-specific loadings with absolute values less than 0.1 were set to 0 before the definition of factor signatures.

## Differential expression analysis of pseudobulk expression profiles

For the acute human heart failure dataset, we performed differential expression analysis across the three profiled conditions per cell type using *edgeR v3.36.0* (*Robinson et al., 2010*) and the pseudobulk filtered data as previously described. Quasi-likelihood negative binomial generalized log-linear models were fitted with edgeR's function glmQLFTest for three different contrasts: myogenic vs fibrotic, myogenic vs ischemic, and fibrotic vs ischemic.

## Functional interpretation of cell-type-specific loadings or factor signatures

To functionally characterize the gene loading matrix associated with a factor of interest, we proposed two alternative enrichment analyses based on the mean expression of MSigDB's hallmarks gene sets (*Liberzon et al., 2015*) and pathway activity footprints (top 500 genes) from PROGENy (*Schubert et al., 2018*). We used *decoupleR's v2.0.1 wmean* function (*Badia-I-Mompel et al., 2022*) to calculate weighted mean scores from factor gene weight matrices to have enrichment scores for each cell type. In the case of PROGENy we used the gene footprints as weights, while for MSigDB's hallmarks we used unweighted means. The normalized scores of each value were calculated with 1000 permutations.

## Estimation of cell-state-dependent gene expression changes upon myocardial infarction captured by the multicellular factor analysis

Given a multicellular program explaining the variance of samples represented by the cell-type-specific gene loadings of a factor, we quantified to what extent it was associated with the emergence of functional cell states of the major cell types analyzed. Our hypothesis was that since each cell-type view summarizes gene expression in the form of pseudobulk profiles per sample, the genes with the highest cell-type-specific absolute weights could be associated with cell states that emerged or increased in a group of samples. We tested this hypothesis in the myocardial infarction dataset by enriching cell-state markers to cell-type-specific factor signatures using hypergeometric tests. Positive and negative signatures were analyzed independently. p-Values were adjusted with the Benjamini-Hochberg

procedure. For these analyses we only included states defined for CMs, Fibs, Endos, and myeloid cells as provided in *Kuppe et al., 2022b*. To calculate cell-state-specific markers, within each cell type, we performed a t-test using *scanpy's v1.9.1 rank_genes_groups* function (*Wolf et al., 2018*) at the single-cell level, contrasting the profiles of all cells belonging to one cell state with the rest of the cells of that cell type. A gene was considered a marker of cell state if the log fold change was greater than or equal to 0.5 and the adjusted p-value less than 0.05.

## Estimation of cell-state-independent gene expression changes upon myocardial infarction captured by the multicellular factor analysis

To quantify the extent to which the multicellular programs captured patient variability that was cell state independent, we assessed whether the expression of a gene, part of a cell-type-specific factor signature, was better explained by sample or cell state variability. We hypothesized that within a cell-type-specific factor signature, it would be possible to find genes with uniform gene expression across distinct functional cell states, which show distinct patterns of expression across distinct groups of samples. This would suggest a general transcriptional shift across cell states. We tested this hypothesis in the myocardial infarction dataset by performing, within each cell type, independent ANOVAs to the expression of each gene belonging to its factor signature. The grouping variable was either the patient condition (myogenic, fibrotic, or ischemic) or the cell state classification. For the former, the ANOVAs were fitted to pseudobulk expression profiles of samples as previously described. For the latter, they were fitted to pseudobulk expression profiles of cell states across samples within each major cell type (CMs, Fibs, Endos, and myeloid cells). Profiles generated with less than 25 cells were excluded in both types of tests. Eta-squared values of the grouping variable per gene were used to quantify the amount of variance explained by cell states or patient conditions. Significance was considered for Benjamini-Hochberg corrected p-values below 0.01. For each gene within each cell-type-specific factor signature, we calculated a log2 ratio between the variance explained by the patient condition and the cell state as a measure of cell-state independence. For this measure, values over 0 represent a greater explained variance associated to the condition rather than the cell state. We performed one-sample t-tests on the distributions of the log ratios of explained variance of each gene for each cell type, to test for general cell state independence across the factor gene signature. Shapiro-Wilk normality tests were performed for the distributions of log ratios of explained variance (adj. p-value <0.01).

## Spatial mapping of cell-type-specific factor signatures

To map cell-type-specific factor signatures to independent ST data from the myocardial infarction dataset, we calculated weighted means of gene expression in each location across all ST slides for the positive and negative signatures separately. Normalized weighted mean scores were calculated with *decoupler-py's v1.1.0 run_wmean* function (*Badia-I-Mompel et al., 2022*) using as weights the gene loadings of each cell-type-specific signature with 100 permutations. Spatial mapping of cell-type factor signatures were only performed in locations where the proportion of the cell type mapped was equal to or greater than 0.1.

To estimate the relative areas across cell types and ST slides where cell-type-specific factor signatures were expressed, we first assumed that the effective area of a signature for a specific cell type was defined by the number of spots where the cell type was present within a ST slide. We consider a cell type to be present in a location if its proportion was equal to or greater than 0.1. Then, for each cell-type-specific factor signature, we counted in how many spots its normalized weighted mean score was greater than two, representing the number of standard deviations from the mean of the distribution of scores from random gene sets. Finally, the relative area of activation of a cell-type-specific factor signature within a slide was calculated as the ratio between the spots with active programs and the effective area.

To visualize the interplay of positive and negative cell-type-specific factor signatures in the ST slides, we encoded the expression of each signature in the red-green-blue (RGB) color space. In this color space, brighter and darker colors represent a high and low expression of a signature respectively and the color combination differentiates different events of co-activation of signatures. To transform the normalized weighted mean estimates into a scale ranging from 0 to 1 so as to be mapped to the

RGB space, each cell-type-specific factor signature was normalized by its maximum value across all slides.

## Multicellular factor analysis for the joint modeling of molecular and tissue-level characteristics of samples

An extended multicellular factor analysis model can be fitted where additional tissue-level characteristics per sample are encoded in views. In the myocardial infarction data, we added four complementary sample views encoding compositional and spatial characteristics of tissues. The first view encoded the compositions of each cell type for each profiled sample. The compositional vector per sample was obtained from *Kuppe et al., 2022b*, where we previously calculated the mean cell-type compositions of the seven analyzed cell types inferred from snRNA, snATAC-seq (chromatin accessibility) data, and ST. The compositional data were transformed to centered-log-ratios using the clr function from the compositions v2.0-4 package (*van den Boogaart and Tolosana-Delgado, 2008*). The other three structural views encode spatial dependencies (i.e. the importance of one cell type in explaining the location and composition of another one in a given tissue) between the seven modeled cell types as inferred by spatially contextualized models as defined by MISTy (*Tanevski et al., 2022*) that we had calculated previously (*Kuppe et al., 2022b*) from ST data. The difference across these spatial views is that each of them model cell-type dependencies in different ranges, the first of them focuses in colocalization of cells within ST locations, the second measures the relationship across cells in immediate neighboring locations (local), and the third one uses an extended effective neighborhood of five spots (extended). The top 21 most variable spatial interactions per view were identified for each view, by sorting the interactions based on the variance of the model's standardized importances. The MOFA model and its interpretation was performed as previously described.

## Generation of patient maps to project and classify new data

To project new patients into a multicellular space inferred by MOFA, we leveraged the feature weights per factor estimated from a reference dataset. In detail, we multiplied the Moore-Penrose generalized inverse matrix (*Penrose and Todd, 1955*) of the concatenated feature weights across views of a reference dataset with the multi-view data of a test cohort of patient samples. To calculate the inverse matrix of the feature weights we used MASS v7.3-57 function ginv(). To classify patient samples into clinical groups we used random forests with a weighted bootstrap sampling process to deal with unbalanced classes using R's *ranger v0.13.1*. For the chronic heart failure atlases we defined a classification task to identify failing (any heart failure etiology) and non-failing (control samples) samples. First, we fitted training models for Chaffin2022 and Reichart2022 independently, using their respective factors inferred with MOFA and sample labels. Then, we projected the patient samples of Chaffin2022 to Reichart2022's factor space and vice versa, as previously described. Then we used the trained random forest to predict the probability of the non-failing class for the newly projected patient samples. Areas under the PRCs were used to evaluate the classification.

## Multicellular factor analysis for the integration of independent cohorts

To generate a multicellular factor analysis that integrates the information of independent patient cohorts, we used MOFA's extension that enables the joint modeling of multiple groups using an extended group-wise prior hierarchy (*Argelaguet et al., 2020*). The main assumption is that the recovered latent space of this group-based analysis will identify factors that explained shared patient variability across studies together with study-specific variability. We fitted a grouped MOFA model to two independent chronic end-stage heart failure single-cell studies to identify multicellular programs that differentiated failing and non-failing heart samples. For each study, we generated pseudobulk normalized expression profiles of cell types for each sample, identified for each cell-type highly variable genes across samples, and filtered out background genes as previously described. Then we selected the collection of highly variable genes per cell type that were shared across studies and used those to create a joint multi-view representation of both datasets. Finally, we fitted a MOFA model as previously described, but with an additional group variable per sample describing the study of origin and an additional feature-level scaling procedure per study.

## Mapping cell-type-specific factor signatures to bulk transcriptomics data

To estimate the expression of cell-type-specific factor signatures in bulk transcriptomics samples, we estimated normalized weighted mean scores per cell-type signature. For a given sample within a bulk transcriptomics study, we calculated normalized weighted mean scores for each cell-type signature using *decoupleR's v2.0.1 wmean* function using as weights their gene loadings with 100 permutations. Before the estimation, gene expression data within a study was centered and scaled across samples.

We calculated the expression of the seven cell-type-specific factor signatures associated with heart failure from the joint multicellular factor analysis model of Chaffin2022 and Reichart2022 for all samples in the 16 RNA-seq and microarray heart failure bulk transcriptomic studies collected in ReHeaT (*Ramirez Flores et al., 2021*). To test for the difference of means of cell-type-specific factor signatures between heart failure and non-failing patients within each study, we used Wilcoxon tests. p-Values were corrected using the Benajmini-Hochberg procedure. Significance was assigned to corrected values lower than or equal to 0.1.

## Benchmarking bulk transcriptomics cell-type deconvolution methods in heart datasets

To evaluate the performance of bulk transcriptomics deconvolution methods in the estimation of cell-type proportions from human heart expression profiles, we benchmarked three methods using the chronic heart failure single-cell datasets. Our benchmark consisted in evaluating the precision of the estimation of cell-type compositions of the samples in Chaffin2022 and Reichart2022 using MuSiC (*Wang et al., 2019*), SCDC (*Dong et al., 2021*), and Bisque (*Jew et al., 2020*) coupled to a healthy heart single-cell reference (*Litviňuková et al., 2020a*; data ref: *Litviňuková et al., 2020b*). First, we selected all apex and left ventricle snRNA-seq samples from the reference study. Then we manually unified cell-type labels and kept all cells belonging to the seven cell types used in the multicellular factor analysis models. Next, for both chronic heart failure single-cell datasets, we created pseudobulk profiles of each patient sample summing up gene counts across all cells from all cell types. As a ground truth, we recorded the real proportions of cell types that were merged into these profiles. We kept the reference and target gene expression matrices in a linear scale and normalized the data using transcripts per million (TPM), as recommended (*Avila Cobos et al., 2020*). Finally, we deconvoluted each pseudobulk sample using MuSiC, SCDC, and Bisque and calculated the Pearson correlation and the root-mean-square error between the estimated and the ground truth cell-type proportions as evaluation metrics.

## Cell-type deconvolution of heart failure transcriptomic datasets

Following the results of the benchmark of cell-type deconvolution methods, we estimated the cell-type proportions of CMs, Fibs, Endos, PCs, vSMCs, and myeloid cells of all samples across seven RNA-seq heart failure bulk transcriptomic studies collected in ReHeaT (*Ramirez Flores et al., 2021*; data ref: *Ramirez Flores et al., 2020*) using SCDC. Each study was TPM normalized and deconvoluted separately. We used Wilcoxon tests to test for the difference of means of cell-type compositions between heart failure and non-failing patients within each study. Compositions were transformed to centered-log-ratios using the *clr* function from the *compositions v2.0-4* package (*van den Boogaart and Tolosana-Delgado, 2008*). p-Values were corrected using the Benajmini-Hochberg procedure. Significance was assigned to corrected p-values lower than or equal to 0.05.

## Comparison of cell-type-specific factor signatures with cell-type proportions for the separation of failing and non-failing hearts from bulk transcriptomics

To evaluate the biological relevance of mapping cell-type-specific factor signatures to bulk transcriptomics, we compared if these signatures were better at distinguishing failing from non-failing hearts than cell-type proportions, estimated from deconvolution methods applied to bulk transcriptomics data. We assumed that the gene expression profile of a bulk transcriptomics sample could be decomposed by the sample's cell-type composition and the disease state of each cell type, as estimated from deconvolution methods and the expression of cell-type-specific factor

signatures, respectively. Silhouette scores of each sample across the seven RNA-seq heart failure bulk transcriptomic studies collected in ReHeaT (*Ramirez Flores et al., 2021*) were calculated relative to their condition (failing and non-failing) from an Euclidean distance matrix calculated either from their cell-type factor signatures or from their estimated cell-type compositions. Compositions were transformed to centered-log-ratios as previously mentioned before calculating the sample distance matrix. Cell-type factor signatures were scaled across all samples from all studies before calculating the sample distance matrix. We performed Wilcoxon tests to compare the silhouette scores for each patient condition. p-Values were corrected using the Benjamini-Hochberg procedure.

## Acknowledgements

RORF and JSR acknowledge the support of DFG through the CRC 1550 'Molecular Circuits of Heart Disease'. JDL and JSR are supported by Informatics for Life funded by the Klaus Tschira Foundation. DD and JSR are supported in part by the European Union's Horizon 2020 research and innovation program (860329 Marie-Curie ITN 'STRATEGY-CKD'). We thank Sebastian Lobentanzer and Olga Ivanova for reading an initial draft of the work and Pau Badia i Mompel, Charlotte Boys, and Robin Fallegger for feedback on the structure of the manuscript. We thank Jovan Tanevski and Ricard Argelaguet for helpful discussions.

## Additional information

### Competing interests

Julio Saez-Rodriguez: reports funding from GSK, Pfizer and Sanofi and fees/honoraria from Travere Therapeutics, Stadapharm, Astex, Pfizer and Grunenthal. The other authors declare that no competing interests exist.

### Funding

| Funder | Grant reference number | Author |
|---|---|---|
| Deutsche Forschungsgemeinschaft | CRC 1550 464424253 | Ricardo Omar Ramirez Flores<br>Julio Saez-Rodriguez |
| Informatics for Life | | Jan David Lanzer<br>Julio Saez-Rodriguez |
| EU ITN Marie Curie Strategy CKD | 860329 | Daniel Dimitrov<br>Julio Saez-Rodriguez |

The funders had no role in study design, data collection and interpretation, or the decision to submit the work for publication.

### Author contributions

Ricardo Omar Ramirez Flores, Conceptualization, Data curation, Software, Formal analysis, Validation, Investigation, Visualization, Methodology, Writing – original draft; Jan David Lanzer, Software, Formal analysis, Investigation, Methodology, Writing – review and editing; Daniel Dimitrov, Software, Investigation, Methodology, Writing – review and editing; Britta Velten, Software, Supervision, Writing – review and editing; Julio Saez-Rodriguez, Resources, Supervision, Funding acquisition, Writing – review and editing

### Author ORCIDs

Ricardo Omar Ramirez Flores ⬮ https://orcid.org/0000-0003-0087-371X
Britta Velten ⬮ https://orcid.org/0000-0002-8397-3515
Julio Saez-Rodriguez ⬮ https://orcid.org/0000-0002-8552-8976

### Decision letter and Author response

Decision letter https://doi.org/10.7554/eLife.93161.sa1
Author response https://doi.org/10.7554/eLife.93161.sa2

# Additional files

## Supplementary files
• MDAR checklist

## Data availability

The datasets and computer code produced in this study are available in the following databases: all scripts related to this manuscript can be consulted at https://github.com/saezlab/MOFAcell (copy archived at *Ramirez Flores, 2023b*); the R package implementing multicellular factor analysis can be found in https://github.com/saezlab/MOFAcellulaR; the python implementation of multicellular factor analysis is available at https://liana-py.readthedocs.io/en/latest/notebooks/mofacellular.html; and a Zenodo entry containing data associated to this manuscript can be accessed at https://zenodo.org/record/8082895.

The following dataset was generated:

| Author(s) | Year | Dataset title | Dataset URL | Database and Identifier |
|-----------|------|---------------|-------------|-------------------------|
| Ramirez Flores RO, Lanzer JD, Dimitrov D, Velten B, Saez-Rodriguez J | 2023 | Multicellular factor analysis of single-cell data for a tissue-centric understanding of disease | https://doi.org/10.5281/zenodo.8082895 | Zenodo, 10.5281/zenodo.8082895 |

The following previously published datasets were used:

| Author(s) | Year | Dataset title | Dataset URL | Database and Identifier |
|-----------|------|---------------|-------------|-------------------------|
| Chaffin M, Papangeli I, Simonson B, Akkad A-D, Hill MC, Arduini A, Fleming SJ, Melanson M, Hayat S, Kost-Alimova M | 2022 | Single-nuclei profiling of human dilated and hypertrophic cardiomyopathy | https://singlecell.broadinstitute.org/single_cell/study/SCP1303/single-nuclei-profiling-of-human-dilated-and-hypertrophic-cardiomyopathy | Broad Single Cell Portal, SCP1303 |
| Kang HM, Subramaniam M, Targ S, Nguyen M, Maliskova L, McCarthy E, Wan E, Wong S, Byrnes L, Lanata CM | 2018 | Multiplexing droplet-based single cell RNA-sequencing using genetic barcodes | https://www.ncbi.nlm.nih.gov/geo/query/acc.cgi?acc=GSE96583 | NCBI Gene Expression Omnibus, GSE96583 |
| Kuppe C, Ramirez Flores RO, Li Z, Hayat S, Levinson RT, Liao X, Hannani MT, Tanevski J, Wünnemann F, Nagai JS | 2022 | Spatial multi-omic map of human myocardial infarction | https://data.humancellatlas.org/explore/projects/e9f36305-d857-44a3-93f0-df4e6007dc97 | Human Cell Atlas Data Portal, e9f36305-d857-44a3-93f0-df4e6007dc97 |
| Litviňuková M, Talavera-López C, Maatz H, Reichart D, Worth CL, Lindberg EL, Kanda M, Polanski K, Heinig M, Lee M | 2020 | Cells of the Adult Heart | https://data.humancellatlas.org/explore/projects/ad98d3cd-26fb-4ee3-99c9-8a2ab085e737 | Human Cell Atlas Data Portal, ad98d3cd-26fb-4ee3-99c9-8a2ab085e737 |
| Reichart D, Lindberg EL, Maatz H, Miranda AMA, Viveiros A, Shvetsov N, Gärtner A, Nadelmann ER, Lee M, Kanemaru K | 2022 | Pathogenic variants damage cell composition and single cell transcription in cardiomyopathies | https://cellxgene.cziscience.com/collections/e75342a8-0f3b-4ec5-8ee1-245a23e0f7cb/private | cellxgene, e75342a8-0f3b-4ec5-8ee1-245a23e0f7cb |

*Continued on next page*

*Continued*

| Author(s) | Year | Dataset title | Dataset URL | Database and Identifier |
|-----------|------|---------------|-------------|-------------------------|
| Ramirez Flores RO, Lanzer JD, Holland CH, Leuschner F, MostPSchultz J-H, Levinson RT, Saez-Rodriguez J | 2020 | The Reference of the Transcriptional Landscape of Human End-Stage Heart Failure | https://doi.org/10.5281/zenodo.3797044 | Zenodo, 10.5281/zenodo.3797044 |

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
