## [Editor Report]

The authors proposed a computational framework, Multicellular Factor Analysis, which is a fundamental advancement in the factor analysis of cross-condition single-cell atlases. The manuscript convincingly demonstrates the application of Multicellular Factor Analysis to uncover multicellular programs associated with disease processes. This innovative framework not only enables unsupervised analysis of single-cell data but also facilitates integration across patient cohorts, marking a helpful contribution to the exploration of molecular alterations in large-scale cross-condition single-cell atlases.

---

## [Decision Letter]

[Editors' note: this paper was reviewed by Review Commons.]

---

## [Author Response]

General Statements [optional]

We have largely revised our manuscript to address the feedback from reviewers, mainly to address the aspects of novelty of our proposed methodology and to clarify the gained biological knowledge from our analyses.

In our manuscript we have proposed the use of multicellular factor analysis to infer multicellular programs from single-cell data and for the unsupervised analysis of samples of cross-condition single-cell and spatial atlases. Our framework repurposes the statistical framework of group factor analysis as implemented in (multi-omics factor analysis) MOFA+. In our revised manuscript, we have emphasized that the novelty of our work is related to the versatile use of a robust statistical framework that complements the analysis toolbox for single-cell analysis, particularly to generate tissue-level descriptions of disease.

In our revisions we have differentiated our work from MOFA+ standard uses, and compared it to what we believe are closer multicellular integration approaches such as DIALOGUE, scITD, and Tensor-cell2cell – which use different computational methods to answer similar biological questions. In our current manuscript, we compared our suggested framework repurposing MOFA+ to DIALOGUE and scITD, and showed the advantages of the flexibility of our approach over these other methods.

The gained biological knowledge from our proposed framework relates to the quantification of cellular molecular coordination in tissues that seems to be independent to the tissue organization and composition. While these results – as those of any method of this sort – remain speculative without follow-up experiments, they are supported by the fact that the findings in the single-cell data sets throughout our manuscript are confirmed in independent datasets of different types (spatial and bulk transcriptomics).

Specifically we have:

Clarified that the novelty of our study is the demonstration of the added value of using group factor analysis as implemented in multi-omics factor analysis (MOFA) to perform unsupervised analysis of samples from cross-condition single-cell data and infer multi-cellular programs over available multicellular integration methods. These advantages relate to the flexibility of MOFA to model missing data, distinct collection of features, and a mixture of tissue-level views thatunifies distinct applications such as the ones presented by scITD and Tensor-cell2cell in a single model,enables a joint model of structural and molecular features of tissues, andfacilitates the integration of multiple studies at the sample level.

All of these features distinguish our work from available methods.

Clarified that the gained biological knowledge relates to the identification of conserved multicellular responses during myocardial remodeling that are independent to the emergence of functional cell-states, cellular compositions and local organization of cells in tissues. These results have not been reported in the publications of the datasets used or elsewhere.Added a new application to peripheral blood mononuclear cells of lupus patients.Added a methodological extension applied to human myocardial infarction data that models not only multicellular programs, but also cell compositions of tissues and spatial organization of cell-types leveraging spatial transcriptomics measurements.Described how multicellular factor analysis can be used to generate patient maps to project novel data to a multicellular space and provide disease classifications.Described the limitations of our study related to cell-cell communications.Provided an R package and a python implementation of multicellular factor analysis in https://github.com/saezlab/MOFAcellulaR and https://liana-py.readthedocs.io/en/latest/notebooks/mofacellular.html, respectively

Point-by-point description of the revisionsReviewer #1 (Evidence, reproducibility and clarity (Required)):Remark to authorsFlores et al. present a pipeline in which they leverage MOFA framework, a matrix factorization algorithm to infer multi-cellular programs (MCPs). Learning and using MCP has already been proposed by others. Yet, authors pursue a similar goals by using MOFA, providing a cell*sample matrix for different cell types as different views (instead of multiple modalities/views) as the input. They later apply MOFA using this data format on a series of applications to analyze acute and chronic human heart failure single-cell datasets using MCPs. Authors further try to expand their analysis by incorporating other modalities.Major points:1.1 As briefly outlined in the remarks, the current manuscript needs novel findings and methodology to grant a research article which I can' see here. The underlying matrix factorization is the original MOFA (literally imported in the code) with no modification to further optimize the method toward the task. While I appreciate and acknowledge the author's efforts resulting in a detailed analysis of heart samples, I think all of these could have been part of MOFA's existing tutorials.

As the reviewer correctly states, we used the framework and code of MOFA. The novelty lies in its application for the unsupervised analysis of samples from cross-condition single-cell data and the inference of multi-cellular programs (MCPs). MOFA is a statistical framework implementing a generalization of group factor analysis with fast inference and its current version fits the task of MCP inference and unsupervised analysis of samples across cell-types that provides a more flexible modeling alternative than current available methods (as presented in Table 1 of the manuscript). Current work on MCP inference is based on the premise of multi-view factorization with distinct statistical modeling alternatives.

Although we show how our work is distinct to MOFA+ standard uses, we believe our work should be put in the context of the other closer multicellular integration approaches: DIALOGUE, scITD, and Tensor-cell2cell. These methods use different computational methods to answer similar biological questions. As mentioned in our general statement, we think that the request to use a novel statistical method to show value, in fact, does not reflect the development in the field. In the same way as DIALOGUE, scITD, and Tensor-cell2cell are not discussed in the context of factorization approaches they repurpose (Penalized matrix decomposition, Tucker decomposition, and CANDECOMP/PARAFAC, respectively), we believe that our work should not be considered in the context of MOFA, nor less innovative simply because we acknowledge the statistical framework that is based upon.

In our revised manuscript we present distinct points that distinguish our discussed methodology from present alternatives and provide evidence about its relevance and uniqueness over available tools:

Simultaneous unsupervised analysis of samples across cell-types and inference of MCPs, together with comprehensive interpretable descriptions of the reconstruction of the original multi-view dataset. This is only currently possible with scITD (Mitchel *et al.,* 2022) and is compared in the manuscript. DIALOGUE (Jerby-Arnon and Regev, 2022) is limited to the generation of MCPs and Tensor-cell2cell (Armingol *et al.,* 2021) is only focused in cell-communications with limited interpretability.Flexible non-overlapping feature set that handles better technical effects such as background expression, as discussed in section “Multicellular factor analysis for an unsupervised analysis of samples in single-cell cohorts”. Moreover, MOFA handles missing data at distinct levels, i.e. samples can miss features within a view, or samples can miss complete views (e.g. as a product of tissue sampling variability), which overcomes the limitations of data completeness that tensor-decomposition-based methods enforce.Joint-modeling of independent atlases that enables meta-analysis at the sample level of cross-condition single-cell data. No currently available methodology is capable of performing similar modeling.Moreover, as mentioned by the reviewer in a later point (Reviewer comment 1.2), MOFA’s modeling flexibility enables joint modeling of distinct aspects of tissues, such as cellular compositions and spatial dependencies. In the new Results section “Multicellular factor analysis for the joint modeling of molecular and tissue-level characteristics of samples” (p. 10), we show how our proposed multicellular factor analysis can be used to model jointly the expression of cell-types, their composition in tissues, and their spatial organization. We used as an example the presented myocardial infarction data where cell compositions estimated from single-nuclei RNA-seq, single-nuclei ATAC-seq, and spatial transcriptomics data were available, together with the quantification of spatial dependencies (i.e. the importance of the location and abundance of a cell-type to predict other cell-types) at different scales. To our knowledge, this analysis alternative is not possible with any of the available methods.Finally, in the Discussion section we describe analysis alternatives and provide vignettes that expand our presented work, including the use of MOFA to model cell-cell communication programs analogous to the work of (Armingol *et al*., 2021) in their Tensor-cell2cell tool (https://liana-py.readthedocs.io/en/latest/notebooks/mofatalk.html).

These differences are also included in Table 1 of our revised manuscript in the columns: “Flexible data type”, “Multiple groups (eg. independent studies)”, and “Handles missing data”.

In sum, our manuscript showed how multicellular factor analysis using MOFA generalizes the applications of (Mitchel *et al.,* 2022), (Jerby-Arnon and Regev, 2022), and (Armingol *et al.,* 2021) in a single model, while expanding to other tissue-level features and meta-analysis of multiple studies.

Finally, we now provide an R package (https://github.com/saezlab/MOFAcellulaR) and python implementations within our analysis framework LIANA (https://liana-py.readthedocs.io/en/latest/notebooks/mofacellular.html) that facilitates the usage of our proposed methodology. The R and python implementations are compatible with current Bioconductor and scverse pipelines, respectively.

Modifications in the manuscript can be seen in Results section “Multicellular factor analysis”

“Compared to other multicellular integration methods tailored for the inference of multicellular programs and sample-level unsupervised analysis of single-cell data (Table 1), multicellular factor analysis using MOFA allows for a more flexible definition of multi-view integration, since it does not restrict cell-type views to the same features. This flexibility enables the inclusion of additional tissue-level descriptions in the model, for example, cell-type compositions, spatial relationships, and cell communication inference scores, representing a generalization of current available methods. MOFA’s structured regularization enables joint-modeling of independent studies making multicellular factor analysis suitable for meta-analysis, a unique feature compared to the aforementioned tissue-centric methods. MOFA’s inference strategy enables multicellular factor analysis to deal with missing data: samples can partially or completely miss cell-type views. MOFA models are computationally efficient (Argelaguet et al., 2020) making multicellular factor analysis scalable to large-scale cross-condition single-cell atlases. The latent space generated with multicellular factor analysis is interpretable, providing measures of the contribution of each view and feature in the construction of the latent space. Finally, building upon these properties, the cell-type specific gene weights can be used to generate patient maps helpful in the projection and classification of new samples, and disease signatures that can be mapped to other modalities such as spatial and bulk omics (Figure 1).“

And in the Discussion section:

“We argue that, in comparison to novel methods explicitly built for the modeling of multicellular responses (Armingol et al., 2022; Mitchel et al., 2022; Jerby-Arnon and Regev, 2022), multicellular factor analysis has three distinct advantages: (1) It enables to better characterize cell-type specific responses and to deal with the technical limitations of cell capture and background noise by not enforcing data completeness across samples and cell-type views, (2) flexible view definition with non-overlapping features that allows for extending the model to include molecular and tissue-level descriptions of tissues, as a generalization of available methods, and (3) joint-modeling of independent studies to generate a shared latent space for samples, which facilitates the integration, comparison and meta-analysis of multiple patient cohorts.”

1.2 How can you explain that the results in donor-level analyses are not due to technical artifacts (batch variation)? Can this be used to infer a new patient similarity map? For example, I would test this by leaving out a few patients from training, projecting them, and seeing where they would end up in the manifold or classifying disease conditions for new patients and explaining the classification by MCPs responsible for that condition.

When knowledge of the technical batches is available it is possible to test for association between these labels and the factors encoding MCPs as shown in Figure 2.

In our current applications, we additionally showed the biological relevance of our estimated MCPs by mapping them to spatial and bulk data sets, which is a direct way of testing how generalizable were our findings:

Regarding the generation of new patient similarity maps, it is possible to estimate the positions of new samples in the manifold formed by the factors representing the MCPs. We have extended our Results section “Multicellular factor analysis for the meta analysis of single-cell atlases of heart failure” to include a description on how patient similarity maps can be used to predict disease condition labels in the context of the two independent patient cohorts of heart failure. Briefly, we fitted random forest classifiers adjusted for unbalanced classes using the factor space inferred for two studies: Reichart2022 and Chaffin2022. Then we projected the samples of one study to the factor space of the other and predicted their condition labels.

Extended section “Multicellular factor analysis for the meta analysis of single-cell atlases of heart failure”:

“Next, we tested if the multicellular programs describing the variability of control and failing myocardium patient samples of each study could be used as reference patient maps where new samples could be projected and classified into a disease condition. First, for each study we generated reference models by training a classifier of healthy and failing myocardium samples from their respective factor scores inferred using random forest (Out of bag prediction error of 0.06 and 0.03 for the model of Reichart2022 and Chaffin2022, respectively). Then, we projected the samples of Reichart2022 into the factor space inferred from the samples of Chaffin2022 and vice versa (Figure 5—figure supplement 1C-D). Finally, we predicted control or failure labels for projected patient samples using the reference classifier and quantified the performance using precision-recall curves (PRCs). The area under the PRC of the classifier of Reichart2022’s patients using Chaffin2022’s factors was 0.69, and we observed a higher performance on the classification of Chaffin2022’s patient samples using Reichart2022’s factors with an area under the PRC of 0.87. These results suggest that the multicellular programs inferred from Reichart2022 better generalize the description of heart failure in comparison to Chaffin2022, which could be explained by the higher degree of variance within the heart failure patients in the former study.**”**

1.3 The bulk and spatial analysis are used posthoc after running MOFA, I think since MOFA can use non-overlapping features set, it would be interesting to see if deconvoluted bulk or ST data can be encoded as another view (one view from scRNAseq data for each cell-type and another view from bulk RNA-seq or ST), you can get normalized expression per spot (for ST) or per sample (for bulk) and use them as input.

Thanks for the suggestion. We agree that the possibility of using non-overlapping features opens options of complex models that include the cell-type compositional and organizational aspects of tissues. However these features must be quantified in the same sample, thus it is limited to samples profiled simultaneously at different scales.

We demonstrate this important modeling definition in the new section “Multicellular factor analysis for the joint modeling of molecular and tissue-level characteristics of samples” and new Figure 4. Here we present the results of a sample-level joint model of multicellular programs together with cell-proportions and spatial dependencies using the myocardial infarction dataset presented in section “Multicellular factor analysis for an unsupervised analysis of samples in single-cell cohorts”. For this dataset based on our previous work we have the compositions of major cell-types and their spatial relationships based on spatially contextualized models (Kuppe *et al.,* 2022).

Modifications in the text were the following:

“Multicellular factor analysis for the joint modeling of molecular and tissue-level characteristics of samples

An additional benefit of performing multicellular factor analysis with MOFA is the flexibility to model distinct views with non-overlapping features that enables the incorporation of other tissue-level characteristics in the unsupervised analysis of samples and inference of multicellular programs, such as cell-type compositions and spatial dependencies (i.e. the importance of a cell-type in predicting the location and abundance of other cell-types) (Figure 4A). This modeling alternative distinguishes multicellular factor analysis from available multicellular program inference methods that are limited to a single molecular aspect of tissues, namely gene expression of cell-types (Mitchel et al., 2022; Jerby-Arnon and Regev, 2022) or cell-communication scores (Armingol et al., 2022).

To showcase the possibility of complementing the inference of multicellular programs with tissue-level descriptions of samples, we extended our previously presented model of human myocardial infarction by including the cell-type compositions of each tissue sample together with spatial dependencies from spatial transcriptomics data inferred with MISTy (Tanevski et al., 2022) (Figure 4B). The extended model incorporated four additional sample views. The first of these new views described the compositions of the seven cell-types analyzed, and the other three views quantified the spatial dependencies between these seven cell-types in three different spatial contexts estimated from spatial transcriptomics: colocalization, local-neighborhood and extended-neighborhood dependencies. The latent space returned by the extended model explained on average 63.8% of the variability of gene expression of the genes across cell-types, showing that the extended model did not lose explanatory power of the molecular views of the tissue samples after adding the structural views since the performance was identical to the original model. The factor scores and gene-weights across cell-types and factors also remained consistent between both models, which could be related to the lower number of features in the additional views. These results suggest that the captured variability of the structural views in the extended model can be related to the coordinated molecular programs associated with myocardial remodeling presented in the past sections.

We observed that the latent space of the extended model captured 70% of the variability in compositions of cell-types of the analyzed tissues and on average 22.8% of the variability in spatial dependencies. Feature weights of Factor 1, which associated the most with the sample condition variables (Kruskal-Wallis test adj. p-value < 0.05), captured expected changes in cell compositions upon myocardial infarction, particularly the difference between control cardiomyocyte-abundant tissues and ischemic immune-abundant ones (Figure 4C). Moreover, the top five highest feature weights across the spatial-dependencies views recovered differential dependencies between immune cells and cells of the vasculature (Figure 4C). The low percentage of explained variance captured by the extended model of the spatial-dependencies views might suggest that the variability in the spatial organization of cells in cardiac tissues can not be mainly explained by the patient conditions, and other variables such as the location of tissue sampling may dominate the signal of spatial transcriptomics. Moreover, the fact that we could identify shared multicellular programs across samples of the same condition despite variable cellular organization suggests a degree of independence between the local organization of cells in cardiac tissues and their overall response to the ischemic context of myocardial infarction. In sum, we have shown how multicellular factor analysis allows us to relate structural characteristics with molecular changes upon disease.”

Materials and methods section:

*“*Multicellular factor analysis for the joint modeling of molecular and tissue-level characteristics of samples

An extended multicellular factor analysis model can be fitted where additional tissue-level characteristics per sample are encoded in views. In the myocardial infarction data, we added four complementary sample views encoding compositional and spatial characteristics of tissues. The first view encoded the compositions of each cell-type for each profiled sample. The compositional vector per sample was obtained from (Kuppe et al., 2022), where we previously calculated the mean cell-type compositions of the seven analyzed cell-types inferred from snRNA, snATAC-seq (Chromatin accessibility) data, and spatial transcriptomics. The compositional data were transformed to centered-log-ratios using the clr function from the compositions v2.0-4 package (van den Boogaart and Tolosana-Delgado, 2008). The other three structural views encode spatial dependencies (i.e. the importance of one cell-type in explaining the location and composition of another one in a given tissue) between the seven modeled cell-types as inferred by spatially contextualized models as defined by MISTy (Tanevski et al., 2022) that we had calculated previously (Kuppe et al., 2022) from spatial transcriptomics data. The difference across these spatial views is that each of them model cell-type dependencies in different ranges, the first of them focuses in colocalization of cells within spatial transcriptomics locations, the second measures the relationship across cells in immediate neighboring locations (local), and the third one uses an extended effective neighborhood of five spots (extended). The top 21 most variable spatial interactions per view were identified for each view, by sorting the interactions based on the variance of the model’s standardized importances. The MOFA model and its interpretation was performed as previously described.

Generation of patient maps to project and classify new data

To project new patients into a multicellular space inferred by MOFA, we leveraged the feature weights per factor estimated from a reference dataset. In detail, we multiplied the Moore-Penrose generalized inverse matrix (Penrose and Todd, 1955) of the concatenated feature weights across views of a reference dataset with the multi-view data of a test cohort of patient samples. To calculate the inverse matrix of the feature weights we used MASS v7.3-57 function ginv(). To classify patient samples into clinical groups we used random forests with a weighted bootstrap sampling process to deal with unbalanced classes. For the chronic heart failure atlases we defined a classification task to identify failing (any heart failure etiology) and non-failing (control samples) samples. First, we fitted training models for Chaffin2022 and Reichart2022 independently, using their respective factors inferred with MOFA and sample labels. Then, we projected the patient samples of Chaffin2022 to Reichart2022’s factor space and vice versa, as previously described. Then we used the trained random forest to predict the probability of the non-failing class for the newly projected patient samples. Areas under the precision-recall curves were used to evaluate the classification.”

Minor:1.4 Some figure references are not correct (e.g., "the single-cell data into a multi-view data representation by estimating pseudo bulk gene expression profiles for each cell-type across samples (Figure 1b)." should be figure 2b)

Thanks for pointing this out. We apologize for these mistakes and we have adjusted all labels correctly.

1.5 The paper is well written, but there could be some more clarifications about what authors consider as cell-type and cell-state, condition, MCPs which I think is critical to current analysis (see here https://linkinghub.elsevier.com/retrieve/pii/S0092867423001599) for the reader not familiar with those concepts.

We agree with the reviewer that it is important to introduce these concepts in more detail to avoid confusion. We have adapted the current manuscript to incorporate these definitions in the introduction and throughout the text.

Modifications in the introduction (p. 2):

“In these studies a common objective is to compare the molecular profiles of cell-types (i.e. cells that potentially share a developmental origin or lineage) across groups of samples (e.g. patient tissues) over distinct conditions or contexts (eg. during disease). Differential gene expression analysis is usually performed for this task, in which the gene expression of each cell-type is contrasted across various conditions (Crowell et al., 2020; Squair et al., 2021). This cell-type-centric approach treats each cell-type-specific alteration in disease independently from each other, ignoring particular gene expression changes of one cell-type that may relate to the changes of other cell-types, here referred to as multicellular programs.”

Definition of cell-states in the Results section “Cell-type-specific factor gene signatures relate to changes in cell-state abundance”:

“We next quantified to what extent the cell-type-specific factor gene signatures recapitulated the emergence of known functional cell-states, here defined as cells within cell-types with distinct functional phenotypes that do not affect their developmental potential (Domcke and Shendure, 2023) (eg. myofibroblasts).”

Reviewer #1 (Significance (Required)):1.6 While I find the concept of MCPs interesting, the current work seems like a series of vignettes and tutorials by simply applying MOFA on different datasets (The authors rightfully state this). However, It needs to be clarified what the novelty is since there is no algorithmic improvement to current MCP methods (because there is no new method) nor novel biological findings. Additionally, even in the current form, the applications are limited to the heart, and the generalization of this proposed analysis pipeline to other tissues and datasets is not explored. Overall, the paper lacks focus and novelty, which is required to grant a publication at this level.

As mentioned in response to 1.1, we show that group factor analysis as implemented in MOFA has advantages over existing tools for the estimation of multicellular programs and unsupervised analysis of samples from single-cell data (see Table 1).

We believe that although the inference of multicellular programs and its use for the unsupervised analysis of samples has been proposed before, in our current work we demonstrate how a well-known statistical framework can be repurposed to circumvent the limitations of current tissue-centric methods. Moreover, we showcase two analysis extensions currently only possible with MOFA:

1) The joint modeling of multicellular programs with other tissue-level views (cell compositions and spatial relationships) and the possibility of adding other descriptions such as cell-communications, which generalizes the ideas presented by independent tools (see new section “Multicellular factor analysis for the joint modeling of molecular and tissue-level characteristics of samples” and new Figure 4, Discussion section, and reviewer response to 1.3).

2) The possibility of jointly modeling independent studies, allowing for meta-analysis of distinct single-cell datasets at the sample level which provides a novel approach of single-cell data integration.

The applications were mainly done in heart data for consistency although they represent four distinct single-cell datasets, one spatial transcriptomics dataset, and 16 independent bulk transcriptomics datasets. That said, as suggested by the reviewer, we additionally show the application of our methodology for the unsupervised analysis of a very different case, peripheral blood mononuclear cell data of lupus patient samples (p. 6; Figure 2—figure supplement 1E).

“To show an additional application of multicellular factor analysis for an exploratory unsupervised analysis of samples profiled with single-cell transcriptomics, we analyzed a peripheral blood mononuclear cell atlas from eight pooled patient lupus samples, each before and after Interferon (IFN)-β stimulation (Kang et al., 2018). After quality control filtering, we analyzed seven cell-types with a median number of highly-variable genes of 459. A model of four factors explained on average 59% of gene expression variability across cell-types. Hierarchical clustering of all factor scores grouped separately stimulated from non-stimulated samples (Figure EV1E). Factor 1, associated with IFN-β stimulation (Kruskall-Wallis test adj. p-value < 0.05), explained on average 50.9% of the variability of gene expression across cell-types, being CD14+ monocytes, FCGR3A+ monocytes, and dendritic cells the cells with the largest explained variance (> 60%), suggesting that these cells may be the most responsive to the stimulation”

expertise: Computational biology, single-cell genomics, machine learningReviewer #2 (Evidence, reproducibility and clarity (Required)):Summary:The authors use MOFA, an unsupervised method to analyze multi-omics data, to create multicellular programs of cross-condition multi-sample studies. First, for each cell-type, a pseudobulk expression matrix per sample is created. The cell-type now functions as the separate view, typically reserved for the different omics layers in MOFA. This then results in a latent space with a certain number of factors across samples. The factors, representing coordinated gene expression changes across cell-types, can then be checked for associations with covariates of interest across the samples.MOFA is well-suited for this task, as it can handle missing data and it is a linear model facilitating the interpretation of the factors. Users should be aware that MOFA can estimate the number of factors, but the pseudobulk profiles require a rigorous selection of cell-type specific marker genes. The result will be most suited for downstream analysis if there is a clear association with one factor and a clinical covariate of interest. In a final step, a positive or negative gene signature can be created by setting a cut-off on the gene weights for that specific factor.The method is applied on 3 separate data sets of heart disease, each time demonstrating that at least one of the factors is associated with a disease covariate of interest. The authors also compare the method to a competitor tool, scITD, and explore to what extent a factor mainly captures variance associated with (i) a general condition covariate or rather (ii) specific cell states.The multicellular programs are also mapped to spatial data with spot resolution. Though this analysis does not bring any novel biological insight in the use case, it does support the claim that the programs are associated with the covariate of interest.The most interesting applications of MOFA are in my opinion the potential for meta-analysis of single-cell studies and validation of cell-type specific gene signatures with publicly available bulkRNAseq data sets.The authors provide various data sets and data types to support their claims and the paper is well written. The relevant code and data has been made available.

We thank the reviewer for the positive comments to our work.

Major comments2.1 What is the added value of the gene signatures obtained from MOFA compared to e.g. a naive univariate approach? In theory, a similar collection of genes or gene signature could be obtained by running a differential gene expression analysis across the samples for each cell-type (e.g. myogenic vs ischemic) and applying a set of relevant cut-offs or filters on the results. In other words, does MOFA detect genes that would otherwise be missed?

Thank you for the relevant comment. The original motivation of our work is the unsupervised analysis of samples based on a manifold formed by a collection of multicellular molecular programs. We envisioned that this unsupervised analysis would be relevant in situations where a clear histological or clinical classification of samples is not possible with reliability. As mentioned by Reviewer #1 in comment 1.2, one advantage of these approaches is that they create patient similarity maps, which have been shown useful to stratify patients in a recent analogous work in multiple sclerosis (Macnair *et al.,* 2022). The cell-type signatures obtained from relevant factors explaining the patient stratification avoid the likelihood of performing “double dipping” by avoiding the need of a direct differential expression analysis between newly formed groups.

In our applications, the generation of cell-type signatures (here called multicellular programs) associated to a specific clinical covariate (eg. control vs perturbation) are post-hoc analyses of the generated manifold. And as the reviewer correctly points out, these signatures should be similar to performing direct differential expression analysis between those patient conditions. In the related work of scITD (Mitchel *et al.,* 2022) the authors showed high concordance between the cell-type signatures and the results of differential expression analysis. We have shown similar results in our expanded section “Multicellular coordinated programs encoded in the latent space”.

“We compared the derived cell-type specific signatures of Factor 1 with traditional differential expression analysis from pseudobulk expression profiles of tissue samples (Figure 3—figure supplement 1C). The median Pearson correlation between the factor gene weights and the log-fold-changes across cell-types was the highest for the contrast between ischemic and myogenic samples (0.98), followed by the contrast between ischemic and fibrotic samples (0.74), and the contrast between fibrotic and myogenic samples (0.65), suggesting that Factor 1 captures the molecular changes associated with the progression of myocardial remodeling, where fibrotic samples represent an intermediary or pseudo-recovered state. Moreover, we observed that from all genes across cell-types included in the multicellular program, 77% of them were differentially expressed (edgeR adj. p-value <= 0.05) in at least one contrast. In summary, our results suggest a high agreement with traditional differential expression testing, with the advantage that the factor scores and gene weights facilitate the analysis of one condition in the context of all the others, avoiding the need to define multiple contrasts.”

It is relevant to mention that in complex experimental designs with multiple conditions, our approach facilitates patient ordering, which allows for the understanding of one condition in the context of all the others, avoiding the need to define multiple contrasts, as mentioned above.

2.2 Could scITD also be used for meta-analysis or could the obtained gene signatures of that method also be mapped to bulkRNAseq data? If so, it would be interesting to show the relative performance with MOFA. If not, this specific advantage should be highlighted.

Thank you for pointing this out. scITD does not provide a group-based model to perform meta-analysis, and this feature is one of the main advantages of group factor analysis as currently implemented in MOFA. We have highlighted this feature in Table 1, the Results section and in the discussion (see reviewer response to 1.1).

Although scITD signatures of a single study could be mapped to bulk transcriptomics data, the stringent tensor representation leads to the generation of signatures that may be influenced by technical effects as shown in the manuscript section ”Multicellular factor analysis for an unsupervised analysis of samples in single-cell cohorts”. Thus we believe that the flexibility of the feature space in MOFA is an advantage for this task.

2.3 Users need to specify gene set signatures based on the weights for a factor of interest. This might suggest a limitation to categorical covariates of interest. If the authors see potential for a continuous covariate of interest, this should at least be highlighted in the text and if possible demonstrated on a use case.

In our applications we limited ourselves to categorical variables, but it is possible to associate factors to continuous variables. An implementation of the association with continuous variables is already available in our newly created R package “MOFAcellulaR”: https://github.com/saezlab/MOFAcellulaR/blob/main/R/get_associations.R.

In the edited manuscript we explicitly mention the possibility of associating factors to continuous and categorical variables in the Results section “Multicellular factor analysis using MOFA”:

“The variables that form this latent space can be interpreted as coordinated transcriptional changes occurring in multiple cells, here referred to as multicellular programs, providing a tissue-centric understanding of the analyzed sample. The inferred multicellular programs can be associated with complementary continuous or categorical variables of the analyzed samples to identify coordinated expression changes related to technical or biological variability.”

Additionally, we demonstrate the possibility of associating factor scores with a continuous covariate of interest in the analysis of Chaffin2022, a chronic heart failure study:

“A mean percentage of explained variance across cell-types of 25% and 21% was associated with heart failure for Chaffin2022 and Reichart2022, respectively (Kruskal-Wallis test adj p-value < 0.05, Figure 5—figure supplement 1A-B). For Chaffin2022, we observed additionally that the left ventricle ejection fraction of the patient samples associated with the same factor describing heart failure, as expected (linear model adj p-value < 0.05, Figure 5—figure supplement 1A).”

Minor comments2.4 In Figure 2c the association between factor 2 and the technical factor shows a very strong outlier. Please verify that the association is still significant after applying a more robust statistical test (e.g. non-parametric test as Wilcoxon).

Thanks for the observation, we have changed the testing to not parametric, with no changes in our conclusion.

2.5 For mapping the cell-type specific factor signatures to bulk transcriptomics, the exact performed comparison or model is unclear. There are seven cell-type signatures for each sample in every study. Was there a t-test run for each cell-type or was a summary measure taken across the cell-types? The thresholding is also rather lenient (adj. p-val 0.1).

We are sorry for not being clear about our procedure, we have included a minor modification in the “Methods and materials” section.

“Mapping cell-type specific factor signatures to bulk transcriptomics data

To estimate the expression of cell-type specific factor signatures in bulk transcriptomics samples, we estimated normalized weighted mean scores per cell-type signature. For a given sample within a bulk transcriptomics study, we calculated normalized weighted mean scores for each cell-type signature using decoupleR’s v2.0.1 wmean function using as weights their gene loadings with 100 permutations. Before the estimation, gene expression data within a study was centered and scaled across samples.”

After identifying the multicellular program associated with heart failure estimated from the two single cell studies meta-analyzed, we calculated the weighted mean expression of the seven cell-type signatures independently to every sample of the 16 bulk studies. In other words each sample within each bulk study will be represented by a vector of 7 values representing the relative expression of a cell-type specific signature (Figure 5D-left). For each bulk transcriptomics study, first, we centered the gene expression data before calculating the weighted mean.

In Figure 5—figure supplement 1G we show the results of performing a Wilcoxon-test of the cell-type scores between heart failure and control samples within each study. Given the relative low sample size of most of the studies (affecting the power of the test), we chose a not so stringent adjusted p-value in the initial manuscript. However, in the revised version we mention the results using a more classical threshold (adj. p-value < 0.05) in this figure.

2.6 typo in abstract: In sum, our framework serves as an exploratory toolfor unsupervised analysis of cross-condition single-cell ***atlas*** and allows for theintegration of the measurements of patient cohorts across distinct data modalities

Thanks for pointing out this typo. We have modified the text.

2.7 In Figure 4a it is not clear to me why on the one hand we see marker enrichment vs loading enrichment with healthy and disease.

We apologize, this is a typo after editing the labels. Both should contain the marker enrichment label. We have fixed this in the new Figure 3—figure supplement 2.

2.8 IN Figure 4b it would help if the same color scheme would be maintained throughout the paper (here now black and white) and if for the cell states the boxplots would be connected per condition, emphasizing the (absence) of change across cell states within a condition.

We thank the reviewer for the suggestion. We have fixed the color scheme in the mentioned panels and connected cell-states within a condition in our new Figure 3—figure supplement 2

Reviewer #2 (Significance (Required)):General assessment:2.9 MOFA is well-suited for detecting multicellular programs because it can handle missing data and allows for easy interpretation of the factors as a linear method. It might have particular potential for meta-analysis across multiple studies and reevaluating bulkRNAseq data sets, but in the current manuscript it is unclear to what extent this is a specific advantage of MOFA or could also be done with competitors. The authors show how the obtained results and associations with clinical covariates can be validated across multiple data types. How the resulting multicellular programs can provide additional biological insight or form the starting point for additional downstream analysis (e.g. cell communication) is not covered in the paper.

We thank the reviewer for highlighting the methodological advantages of group factor analysis for the estimation of multicellular programs and the unsupervised analysis of samples from cross-condition single-cell atlas. As mentioned in response to 1.1 and 2.2, the added value of our methodology is the flexibility of feature views (that goes beyond gene expression) and simultaneous modeling of independent single-cell datasets, a feature not present in any of the currently available methods that facilitates the meta-analysis of datasets across modalities.

While we interpret the presented multicellular programs in the context of cellular functions and the division of labor of cell states, it is true that we did not attempt to provide mechanistic hypotheses, for example, via cell-cell communication, on how this coordination across cell-types emerges. We highlight this topic as a further direction in our Discussion section:

“While our proposed approach enables the inference of tissue-level coordinated responses across cell-types in distinct contexts, the connection of these processes to cell-cell communication events remains an open challenge. Applications of group factor analysis with MOFA including views measuring the co-expression of ligands and receptors from pairs or groups of cells to infer cell-cell communication programs are possible, analogous to the work of (Armingol et al., 2022; Baghdassarian et al., 2023), as shown in the tutorials of our cell-cell communication tool LIANA^+^ (Dimitrov et al. 2023) (https://liana-py.readthedocs.io/en/latest/notebooks/mofatalk.html). Alternatively, the estimation of multicellular programs could be further used to inform the inference of mechanistic network models connecting inter- and intra-cellular signaling events. However, these approaches are limited by the potential of transcriptomics measurements in explaining cell-cell communication.”

Audience: This paper will be mainly of interest to a specialized public interested in unsupervised methods for large scale multi-sample and multi-condition studies.Reviewer: main background in the analysis of scRNAseq data.Reviewer #3 (Evidence, reproducibility and clarity (Required)):This manuscript by Saez-Rodriguez and colleagues proposes to repurpose Multi-Omics Factor Analysis for the use of single cell data. The initial open problem stated by the paper is the need for a framework to map multicellular programs (such as derived from factor analysis) to other modalities such as spatial or bulk data. The authors propose to repurpose MOFA for use in single cell data. Case studies involve human heart failure datasets (and focuses on spatial and bulk comparisons).There are particular issues with clarity regarding the key methodological contribution (and assessment of it), discussed under significance.Reviewer #3 (Significance (Required)):3.1 I am very puzzled by the repeated claims the manuscript makes that their central methodological contribution and innovation is to use MOFA for single cell data. One of their citations for MOFA is to MOFA+, which is precisely that (in a relatively popular manuscript published by the original authors of MOFA and not overlapping with the present authors). I am left to wonder what I missed.

We apologize for the misunderstanding, as mentioned in the response to 1.1 and explained by reviewer 2’s summary, the main objective of our work is to use the statistical framework of group factor analysis for the inference of multicellular programs and the sample-level unsupervised analysis of cross-condition single-cell data, which is a distinct task to multimodal integration (Argelaguet *et al.,* 2021).

While it is true that MOFA+ introduced expansions to the model for the modeling of single-cell data, namely fast inference and group-based modeling, the main focus in their applications is the multimodal integration of data, where each cell is represented by a collection of distinct collection of features (e.g. chromatin accessibility and gene expression). Unlike multimodal integration, here we propose a different approach to analyze single-cell data at the sample level instead of the cell level, without modifying the underlying statistical model (see section 2.1 of the manuscript).

In detail, what we assume is that samples of single-cell transcriptomics data (e.g. tissue from a patient) can be represented by a collection of independent vectors collecting the gene expression information of cell types composing the tissue analyzed. Decomposition of these multiple views with group factor analysis produces a manifold that captures multicellular programs (coordinated expression processes across cell-types), or shared variability across cell-types simultaneously. Altogether, this represents a novel usage of group factor analysis in an application for the inference of multicellular programs, where the main focus is not at the cell-level but at the patient level.

As a side note, Britta Velten, one of main developers of MOFA and coauthor of both the MOFA and MOFA+ papers, is a contributor and coauthor of this manuscript, and Ricard Argelaguet, who also led both versions of MOFA, gave us helpful feedback and is acknowledged as such on this work.

3.2 Multimodal integration methods are fairly numerous and even if they're not all exactly factor analyses, it's strange to argue that MOFA fills some unique conceptual gap. I agree it fills something of an interesting gap (except for MOFA+ already filling it), but it's not like the quite popular spatial to single-cell integration approaches aren't doing similar things. If this is a methods paper (as it is presented) then there would have to be very substantially more comparative evaluation to these other approaches.

As presented in the previous response (3.1) our current work is not focused on multimodal integration, but rather the inference of multicellular programs and the sample-level unsupervised analysis of single-cell data. Given this, in the current manuscript we compared our proposed methodology with the only three other available methods that address at least partially the inference of multicellular programs (see Table 1 in our manuscript). In response to 1.1 and 3.2 we discussed the advantages of our proposed methodology compared to available methods. In the manuscript section 2.2 we compared group factor analysis with tensor decomposition and showed that the former better deals with technical artifacts and better identifies known patient groups.

We have distinguished our work from multimodal integration explicitly in the introduction of our revised manuscript (p. 2).

“A set of novel tissue-centric computational methods for multicellular integration have emerged that are helpful in the definition of multicellular programs associated with clinical covariates of interest (Jerby-Arnon and Regev, 2022), and the unsupervised analysis of samples from cross-condition single-cell atlases (Armingol et al., 2022; Mitchel et al., 2022). These multicellular integration methods are extensions of matrix factorization that aim to reduce the dimensionality of the data while retaining most of the variability. In contrast to classic approaches, such as principal component analysis, these methods are capable of dealing with higher order representations, such as the ones from single-cell data, where a sample is described by a collection of different cell-types. A key element of multicellular integration methods is that they first transform cross-condition single-cell data into a multi-view representation, in which each view contains the summarized gene expression profile across cells of the same type for each sample. Unlike multimodal integration, where each cell is represented by a collection of distinct feature modalities (e.g. chromatin accessibility and gene expression) and the objective is to map the features across modalities, in multicellular integration the objective is to measure the variability of samples (eg. patient tissues) across multiple cell-types simultaneously.”

3.3 The biological use cases are comparatively interesting and dominate the manuscript (but are still presented principally as use cases rather than a compelling biological narrative of their own).

The focus of our manuscript was the introduction of group factor analysis for the novel applications of the inference of multicellular programs and the sample-level unsupervised analysis from single-cell data. Given the distinct possibilities of post-hoc analyses, we mainly used acute and chronic heart failure data to showcase the utility of MOFA to connect spatial and bulk modalities with single-cell data.

That said, our analyses allowed to generate novel hypotheses of these data sets regarding the activation of multicellular programs during disease that are independent to the local composition and organization of cell-types:

We summarize these results in the third paragraph of the discussion in the manuscript (p. 15, 16):

“In an application to a collection of public single-cell atlases of acute and chronic heart failure, we found evidence of dominant cell-state independent transcriptional deregulation of cell-types upon myocardial infarction not found by previous analyses. This may suggest that while certain functional states within a cell-type are more favored in a disease context, most of the cells of a specific type have a shared transcriptional profile in disease tissues. If part of this shared transcriptional profile is interpreted as a signature of the tissue microenvironment that drives cells in tissues towards specific functions, this result may also indicate that a major source of variability across tissues, besides cellular composition, is the degree in which the homeostatic transcriptional balance of the tissue is disturbed. By combining the results of multicellular factor analysis with spatial transcriptomics datasets, we explored this hypothesis and identified larger areas of cell-type-specific transcriptional alterations in diseased tissues. Moreover, extending our multicellular factor analysis model with the spatial relationships across cell-types revealed a degree of independence between the activation of myocardial remodeling programs and the local organization of cells in the tissue, a finding not reported in the original manuscript of the analyzed dataset or elsewhere. Given these observations on global alterations upon myocardial infarction, we meta-analyzed single-cell samples from two additional studies of healthy and heart failure patients with multiple cardiomyopathies. Here, we found a conserved transcriptional response across cell-types in failing hearts, despite technical and clinical variability between patients. Further, we could find traces of these cell-type alterations in bulk data sets that were independent to the cellular compositions of tissues. These observations suggest that our approach can estimate cell-type-specific transcriptional changes from bulk data that, together with changes in cell-type compositions, describe tissue pathophysiology. Altogether, these results highlight how multicellular factor analysis can be used to integrate the measurements of independent single-cell, spatial, and bulk datasets to measure cell-type alterations in disease.”

To fully assess the relevance of these observations, they should be investigated in more datasets and analyses, where shared functional cell-states across distinct heart failure etiologies are identified and then compared at their compositional and molecular level. This, in our opinion, represents an independent study on its own.

3.4 Altogether, I found the framing of this manuscript very puzzling. It is possible the result would be more clearly presented if the use case was the major focus rather than the more conceptual point about factor analysis.

Thanks for the suggestion. The major aim of this manuscript is to highlight the versatility of the generalization of group factor analysis as implemented in MOFA for novel applications in single-cell data analysis, beyond multimodal integration of single cells. The definition of multicellular programs from single-cell data and its sample-level unsupervised analysis are relatively new analyses in the field, and thus we believe that it is timely to show how a known statistical framework can be used for these applications.

We believe that a detailed analysis of single-cell datasets of heart failure deserves its own focus and it is out of scope of our current objective with this manuscript. We apologize for the apparent misunderstanding of the objective of our methodology. As mentioned in response to 3.2 we have added these distinctions in the introduction of the manuscript.

References

Argelaguet R, Cuomo ASE, Stegle O and Marioni JC (2021) Computational principles and challenges in single-cell data integration. *Nat Biotechnol* 39: 1202–1215

Armingol E, Baghdassarian H, Martino C, Perez-Lopez A, Knight R and Lewis NE (2021) Context-aware deconvolution of cell-cell communication with Tensor-cell2cell. *BioRxiv*

Jerby-Arnon L and Regev A (2022) DIALOGUE maps multicellular programs in tissue from single-cell or spatial transcriptomics data. *Nat Biotechnol* 40: 1467–1477

Kuppe C, Ramirez Flores RO, Li Z, Hayat S, Levinson RT, Liao X, Hannani MT, Tanevski J, Wünnemann F, Nagai JS, *et al.* (2022) Spatial multi-omic map of human myocardial infarction. *Nature* 608: 766–777

Macnair W, Calini D, Agirre E, Bryois J, Jaekel S, Kukanja P, Stokar-Regenscheit N, Ott V, Foo LC, Collin L, *et al.* (2022) Single nuclei RNAseq stratifies multiple sclerosis patients into three distinct white matter glia responses. *BioRxiv*

Mitchel J, Gordon MG, Perez RK, Biederstedt E, Bueno R, Ye CJ and Kharchenko P (2022) Tensor decomposition reveals coordinated multicellular patterns of transcriptional variation that distinguish and stratify disease individuals. *BioRxiv*